# Record high room temperature resistance switching in ferroelectric-gated Mott transistors unlocked by interfacial charge engineering

Yifei Hao [1,7], Xuegang Chen[1,7], Le Zhang[1], Myung-Geun Han [2], Wei Wang [2], Yue-Wen Fang [3,4], Hanghui Chen [5,6], Yimei Zhu[2] & Xia Hong [1] ✉

The superior size and power scaling potential of ferroelectric-gated Mott transistors makes them promising building blocks for developing energy-efficient memory and logic applications in the post-Moore's Law era. The close to metallic carrier density in the Mott channel, however, imposes the bottle-neck for achieving substantial field effect modulation via a solid-state gate. Previous studies have focused on optimizing the thickness, charge mobility, and carrier density of single-layer correlated channels, which have only led to moderate resistance switching at room temperature. Here, we report a record high nonvolatile resistance switching ratio of 38,440% at 300 K in a prototype Mott transistor consisting of a ferroelectric $PbZr_{0.2}Ti_{0.8}O_3$ gate and an $R$NiO$_3$ ($R$: rare earth)/La$_{0.67}$Sr$_{0.33}$MnO$_3$ composite channel. The ultrathin La$_{0.67}$Sr$_{0.33}$MnO$_3$ buffer layer not only tailors the carrier density profile in $R$NiO$_3$ through interfacial charge transfer, as corroborated by first-principles calculations, but also provides an extended screening layer that reduces the depolarization effect in the ferroelectric gate. Our study points to an effective material strategy for the functional design of complex oxide heterointerfaces that harnesses the competing roles of charge in field effect screening and ferroelectric depolarization effects.

The device concept of a ferroelectric field effect transistor (FeFET) built upon a Mott channel has been extensively investigated over the last three decades for its potential in developing nonvolatile memory and energy-efficient logic applications that can outperform the CMOS technology[1–3]. Perovskite oxide heterostructures emerge as the most promising material candidates, where atomically sharp interfaces with low defect densities can be achieved due to the structural similarity between functional distinct oxide layers[2–4]. Capitalizing on the inter-facial coupling between lattice[5,6], orbital[7–9], charge[9–14], and spin[15] degrees of freedom, the ferroelectric/correlated oxide hetero-structures can host various emergent phenomena that are not acces-sible in bulk, including nonvolatile voltage control of magnetic order[4,16] and magnetic anisotropy[9,17], polarization tuning of topological spin textures[18] and superconductivity[19,20], and domain wall enabled

[1]Department of Physics and Astronomy & Nebraska Center for Materials and Nanoscience, University of Nebraska-Lincoln, Lincoln, NE 68588-0299, USA. [2]Condensed Matter Physics and Materials Science, Brookhaven National Laboratory, Upton, NY 11973-5000, USA. [3]Fisika Aplikatua Saila, Gipuzkoako Inge-niaritza Eskola, University of the Basque Country (UPV/EHU), Europa Plaza 1, 20018 Donostia/San Sebastián, Spain. [4]Centro de Física de Materiales (CSIC-UPV/EHU), Manuel de Lardizabal pasealekua 5, 20018 Donostia/San San Sebastián, Spain. [5]NYU-ECNU Institute of Physics, NYU Shanghai, 200062 Shanghai, China. [6]Department of Physics, New York University, New York, NY 10002, USA. [7]These authors contributed equally: Yifei Hao, Xuegang Chen. ✉e-mail: xia.hong@unl.edu

**Fig. 1 | Sample characterization. a** Device schematic. **b** XRD $\theta$ $-2\theta$ scan taken on a PZT/NNO(4)/LSMO(5) heterostructure deposited on (001) SrTiO₃ (STO) substrate. Inset: AFM topography image of the sample. **c** Cross-sectional HRSTEM image and **d**, EELS element mapping taken on a PZT/LNO(3)/LSMO(3) sample. **e**–**h** PFM characterizations and resistance switching taken on a PZT/LNO(4)/LSMO(2) sample. **e** PFM phase and **f** amplitude images of concentric square domains. **g** PFM phase and amplitude switching hysteresis. **h** $R_{\square}$ vs. $V_g$ at 300 K.

tunneling effects[21]. Compared to other nano-materials, complex oxides are an appealing platform for the technological implementation of Mott FeFETs due to the scalability of epitaxial thin film growth and potential to be integrated with conventional semiconductors[22,23].

The key advantage of developing Mott transistors is the close-to-metallic carrier density of strongly correlated materials ($10^{21}$–$10^{23}$/cm³), which corresponds to a sub-nanometer scale screening length[24] that can transcend the fundamental size scaling limit of conventional semiconductors[25]. This attribute, however, also imposes a major bottleneck on the magnitude of the field effect, as the doping level required to induce a substantial channel resistance modulation well exceeds what can be achieved via a solid-state gate. On the other hand, charge screening provided by the conductive channel is essential for minimizing the depolarization field, reducing domain formation, and even sustaining ferroelectricity in the ferroelectric gate[26,27]. Finding an effective material strategy to harness the multi-faceted roles of charge in the Mott channel is thus imperative for designing devices with optimal size scaling, resistance on-off ratio, and retention behavior.

Previous studies of Mott FeFETs have focused on optimizing single-layer Mott channels, either working with channel materials with intrinsically low carrier density[14] or engineering the thickness[13,24] and charge mobility[5,7,13] of the correlated oxides. These efforts, however, only yield moderate enhancement of the ferroelectric field effect due to the competing roles of charge. The former approach corresponds to a large depletion width, losing the competitive edge of this device concept[14]. For the latter, the highest room temperature resistance on-off ratio reported to date is 11.4, which has been achieved in 3.5 unit cell (uc) termination-controlled LaNiO₃ channel[7]. Further reducing the channel thickness results in an insulating or electrically dead layer, a phenomenon widely observed in correlated oxides[24,28–33], where the strong depolarization field compromises nonvolatile field effect switching due to the absence of itinerant screening charges. The central challenge for designing the single-layer Mott channel is to satisfy the conflicting requirements of reducing the effective sheet

carrier density while preserving effective screening of the depolarization field in the ferroelectric gate.

In this work, we exploit ferroelectric PbZr₀.₂Ti₀.₈O₃ (PZT)-gated rare earth nickelates $R$NiO₃ ($R$ = La, Nd, Sm) as a model system to show that a record high nonvolatile resistance switching ratio of 38,440% can be achieved at 300 K by inserting an ultrathin charge transfer layer La₀.₆₇Sr₀.₃₃MnO₃ (LSMO). A systematic comparison between $R$NiO₃ single-layer channels and $R$NiO₃/LSMO bilayer channels (Fig. 1a) shows that the resistance switching ratio increases exponentially with decreasing channel thickness till the channel becomes electrically dead, while the composite channels with the same total channel thickness ($t_{tot}$) exhibit up to three orders of magnitude higher resistance modulation. The drastically enhanced field effect has been attributed to a tailored carrier density profile in the $R$NiO₃ channel due to the interfacial charge transfer with LSMO, as corroborated by density functional theory (DFT) calculations. Our study reveals the intricate interplay between the field effect doping in $R$NiO₃, interfacial charge transfer between $R$NiO₃ and LSMO, and finite screening length induced depolarization effect in the ferroelectric gate, pointing to an effective route for building high density, low power nanoelectronics and spintronics via functional complex oxide heterointerfaces.

## Results and discussion
### Characterization of oxide heterostructures
We work with the rare earth nickelates as the Mott channel. For $R$NiO₃, reducing the rare earth cation size (i.e., La → Nd → Sm) leads to enhanced lattice distortion, suppressed charge itinerancy, and thus higher metal-insulator transition temperature ($T_{MI}$)[34]. Bulk SmNiO₃ and NdNiO₃ are the charge-transfer-type Mott insulators, while LaNiO₃ is a correlated metal. Epitaxial LaNiO₃ films exhibit a film thickness-driven metal-insulator transition upon the dimensionality-crossover[28]. Previous theoretical studies have predicted that the surface layer of ultrathin LaNiO₃ is an orbital-specific Mott insulator[32]. Ab initio calculations also show that the carrier density of ultrathin LaNiO₃ does not

scale linearly with film thickness but rather drops abruptly below 5 uc[31], suggesting that it approaches the correlation gap.

We deposit epitaxial PZT/$R$NiO$_3$ and PZT/$R$NiO$_3$/LSMO heterostructures on mixed termination (001) LaAlO$_3$ and SrTiO$_3$ substrates, respectively ("Methods"). Atomic force microscopy (AFM) measurements reveal smooth surfaces with a typical root-mean-square roughness of about 5 Å (Fig. 1b inset). X-ray diffraction (XRD) measurements show (001) growth of all layers (Fig. 1b and Supplementary Fig. 1). The high crystallinity of the samples is confirmed by high-resolution scanning transmission electron microscopy (HRSTEM) measurements (Methods), which reveal atomically sharp interfaces between the constituent layers (Fig. 1c and Supplementary Fig. 2). Figure 1d shows the element mapping of O, Ti, Mn, La, Sr, and Pb taken on a PZT/3 uc LaNiO$_3$ [LNO(3)]/3 uc LSMO [LSMO(3)] sample using electron energy loss spectroscopy (EELS). The interfaces are chemically sharp, and the interfacial intermixing is limited within the unit cell level. The samples are patterned into FET devices with Hall-bar geometry for transport studies (Methods).

Figure 1e, f shows the piezoresponse force microscopy (PFM) images of polarization up ($P_{up}$) and down ($P_{down}$) domains written on a PZT/LNO(4)/LSMO(2) sample. In the as-grown state, PZT is uniformly polarized in the $P_{up}$ state (Fig. 1e, f). The PFM switching hysteresis loop reveals coercive voltages of about −1.0 V for the $P_{up}$ state and +2.9 V for the $P_{down}$ state (Fig. 1g), consistent with the preference for the $P_{up}$ state. Such polarization asymmetry has been widely observed in epitaxial PZT films deposited on correlated oxide electrodes[9,17], which can be attributed to the asymmetric electrical boundary conditions. Figure 1h shows the sheet resistance $R_\square$ as a function of gate voltage $V_g$ for this sample at 300 K, with the switching voltages consistent with the PFM result. The $P_{up}$ ($P_{down}$) polarization corresponds to the enhanced (suppressed) channel conduction, denoted as the $R_{on}$ ($R_{off}$) state, agreeing with the $p$-type doping for LaNiO$_3$[7].

## Single-layer $R$NiO$_3$ channels

Previous studies show that $R$NiO$_3$ thin films possess more than one hole per unit cell[5,11,29,31]. The sheet carrier density $n_\square$ for a $R$NiO$_3$ channel of a few nanometers thickness can well exceed the bound charge density of PZT[5]. Scaling down the channel thickness can

effectively reduce $n_\square$, but its boost in the field effect modulation cannot be sustained below the electric dead layer thickness, where the emergence of a correlation gap or interface/surface-related disorder renders an insulating state[24,28–33], causing strong depolarization. To identify the optimal channel thickness, we first characterize the film-thickness dependence of conduction in LaNiO$_3$, NdNiO$_3$ (NNO), and SmNiO$_3$ (SNO) films (Fig. 2a–d). For LaNiO$_3$ (Fig. 2a), $R_\square(T)$ shows metallic behavior ($\frac{dR}{dT} > 0$) in films down to 2.5 uc (about 1 nm) thickness. For the 4 uc and thinner films, a slight resistance upturn is observed at low temperature, which can be attributed to weak localization or enhanced electron correlation accompanied with dimensionality crossover[28,33]. The 2 uc film, on the other hand, exhibits insulating behavior over the entire temperature range. The critical thickness for the metal-insulator transition is consistent with previous reports[13,31,32]. Similar thickness dependence of $R_\square(T)$ is observed in NdNiO$_3$ (Fig. 2b). Due to the compressive strain on LaAlO$_3$ substrates, the 22 uc NdNiO$_3$ film remains metallic down to 10 K[30]. The low-temperature insulating phase emerges in the 15 uc and thinner films, and the electric dead layer thickness is about 4 uc. For both LaNiO$_3$ and NdNiO$_3$, the transition to insulating behavior occurs as $R_\square$ exceeds the two-dimensional quantum resistance $\frac{h}{2e^2} \sim 12.9$ kΩ, suggesting the emergence of strong localization[28,33]. In contrast, thick SmNiO$_3$ films (8–26 uc) exhibit insulating behavior below 320 K even with $R_\square$ well below $\frac{h}{2e^2}$ (Fig. 2c), indicating it is intrinsic to SmNiO$_3$. Accompanied with its highly distorted lattice, bulk SmNiO$_3$ possesses a $T_{MI}$ of about 400 K[34]. Empirically, the room temperature resistivity of $R$NiO$_3$ films of the same thickness follows $\rho_{SNO} > \rho_{NNO} > \rho_{LNO}$, and the samples become insulating when resistivity exceeds ~1 mΩ cm (Fig. 2d). It is expected that the more conductive LaNiO$_3$ and NdNiO$_3$ can sustain further channel thickness scaling for the field effect studies.

Next, we investigate the ferroelectric field effect in single-layer $R$NiO$_3$ channels (Fig. 2e). Figure 3a shows $R_\square(T)$ taken on a PZT/4 uc LaNiO$_3$ sample. In the $P_{up}$ state, the sample exhibits metallic behavior at high temperature followed by a resistance upturn at 70 K. By switching the polarization to the $P_{down}$ state, the sample becomes insulating at room temperature, signaling a carrier density-driven Mott transition. Hall effect measurements taken on this sample reveal hole-type doping and $n_\square$ of $8.04 \times 10^{15}$ cm$^{-2}$ for the on state and $7.10 \times 10^{15}$ cm$^{-2}$ for the off state (Supplementary Note 2), confirming

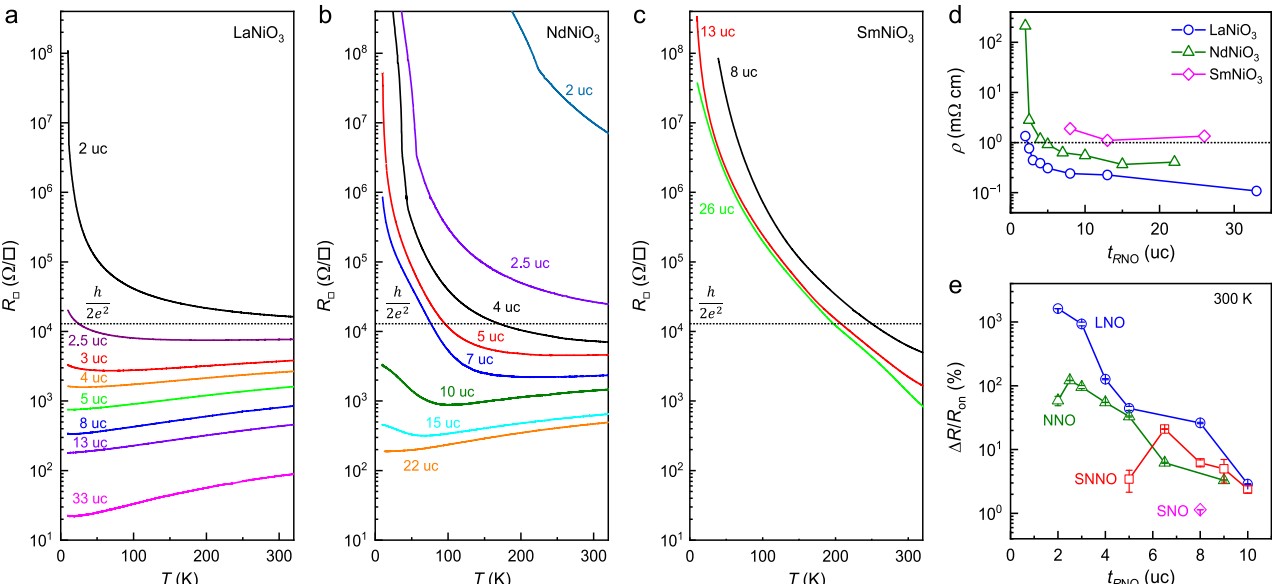

**Fig. 2 | Thickness dependence of conduction and field effect in single-layer $R$NiO$_3$.** **a**–**c** $R_\square(T)$ of **a** LaNiO$_3$, **b** NdNiO$_3$, and **c** SmNiO$_3$ films with various thicknesses deposited on LaAlO$_3$ substrates. **d** Resistivity $\rho$ vs. film thickness at 300 K for

$R$NiO$_3$ films. **e** $\Delta R/R_{on}$ vs. channel thickness at 300 K for single-layer $R$NiO$_3$ channels (error bars: SD).

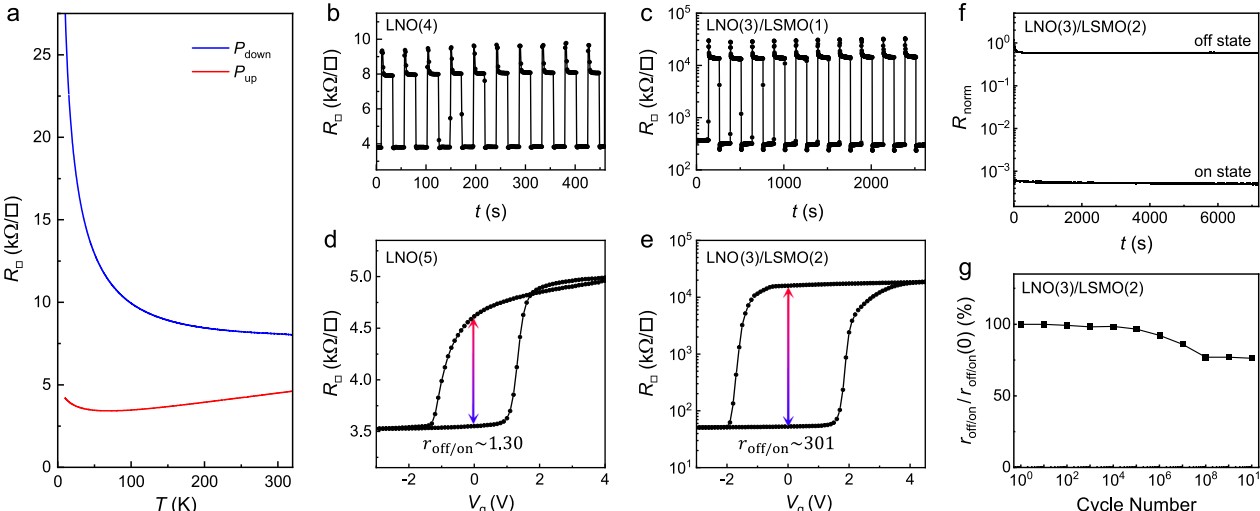

**Fig. 3 | Ferroelectric field effect at 300 K in single-layer and bilayer channels.** **a** $R_{\square}(T)$ taken on a single-layer 4 uc LaNiO$_3$ channel for both polarization states of PZT. **b, c** $R_{\square}$ switching upon application of voltage pulses taken on devices with **b** single-layer LNO(4) and **c** bilayer LNO(3)/LSMO(1) channels. **d, e** $R_{\square}$ vs. $V_g$ switching hysteresis taken on devices with **d** single-layer LNO(5) and **e** bilayer LNO(3)/LSMO(2) channels. **f** Time dependence of the normalized on- and off-state resistance, and **g** normalized $r_{\text{off/on}}$ vs. cycle number taken on a PZT/LNO(3)/LSMO(2) sample. The initial on-state value (defined as 0) is not shown on the log scale in (**f**).

the nonvolatile ferroelectric polarization doping. The on-state carrier density is comparable with that of single-layer LaNiO$_3$ (Supplementary Fig. 6) and similar to those in NdNiO$_3$[11] and Sm$_{0.5}$Nd$_{0.5}$NiO$_3$[29] films. From $2P_r = n_{\square}e$, we deduce the remnant polarization $P_r$ of PZT to be about 76 μC cm$^{-2}$, which is in good agreement with the $P$-$E$ loop measurement taken on a PZT/10 nm LaNiO$_3$ capacitor device (~77 μC cm$^{-2}$) (Supplementary Fig. 4) and the Hall results obtained on PZT/Sm$_{0.5}$Nd$_{0.5}$NiO$_3$[5].

We also note that depositing the PZT top layer suppresses the channel conductivity. The on-state $R_{\square}$ of PZT-gated single-layer $R$NiO$_3$ channels is always higher than that of the uncapped film with the same thickness. Similar suppression in conduction has also been observed in the NdNiO$_3$ channels. Such changes may share a mechanism similar to that observed in ref. 35, where a close lattice-matched capping layer LaAlO$_3$ relieves the lattice distortion induced by surface reconstruction, enhancing LaNiO$_3$ conductivity. The PZT capping layer, in contrast, imposes a large tensile strain, suppressing the $R$NiO$_3$ conduction[29,30].

The intricate relation between the field effect screening and charge itinerancy is clearly manifested in the resistance modulation. We characterize the room temperature resistance switching ratio upon polarization switching of PZT, defined as $\Delta R/R_{\text{on}} = (R_{\text{off}} - R_{\text{on}})/R_{\text{on}}$, in $R$NiO$_3$ devices with various channel thickness $t_{R\text{NO}}$ (Supplementary Figs. 7–9). Figure 2e shows $\Delta R/R_{\text{on}}$ vs. $t_{R\text{NO}}$ at 300 K for the single-layer channels. For a systematic comparison, we also include the data for Sm$_{0.5}$Nd$_{0.5}$NiO$_3$ channels reported in a previous study[12]. Despite the difference in charge mobility and carrier density, LaNiO$_3$, NdNiO$_3$, and Sm$_{0.5}$Nd$_{0.5}$NiO$_3$ show highly consistent scaling behavior of the field effect: $\Delta R/R_{\text{on}}$ increases exponentially with decreasing $t_{R\text{NO}}$ in the metallic phase, peaking around the electric dead layer thickness; further reducing the channel thickness yields a sharp suppression of the field effect. The initial increase in $\Delta R/R_{\text{on}}$ reflects the interfacial nature of the charge screening effect, which is the dominant mechanism underlying the resistance modulation[24,36]. Note that we do not consider the effect of interface termination on charge mobility, as all samples are prepared on substrates with mixed termination[7,12]. The diminished field effect in ultrathin channels below the dead layer thickness can be attributed to the incomplete screening of the depolarization field in PZT, which suppresses the switchable polarization and compromises the retention behavior[5,26,27]. This scenario also explains the small resistance modulation observed in the insulating SmNiO$_3$ devices

($\Delta R/R_{\text{on}}$ ~ 1.14% for the 8 uc channel). The highest resistance switching ratios observed in LaNiO$_3$ (NdNiO$_3$) is about 1,619% (121%) for the 2 uc (2.5 uc) channel, well exceeding the peak $\Delta R/R_{\text{on}}$ of Sm$_{0.5}$Nd$_{0.5}$NiO$_3$ (21% for the 6.5 uc channel)[12]. As expected, the largest field effect is observed in the most conductive material (LaNiO$_3$), which has the lowest electric dead layer thickness (2 uc).

## Enhanced ferroelectric field effect in $R$NiO$_3$/LSMO bilayer channels

For the 2 uc LaNiO$_3$ channel, the resistance on/off ratio is $r_{\text{off/on}} = \frac{R_{\text{off}}}{R_{\text{on}}} = 17.2$, the highest room temperature value for Mott FeFETs with single-layer channels, but well below the expected tuning effect for a Mott transition. For this channel geometry, the key challenge for optimizing the field effect is imposed by the competing roles of charge in field effect modulation and polarization screening: reducing the channel thickness can effectively reduce the net carrier density, but it also leads to a truncated screening region and renders partial screening of depolarization field. A possible solution to decouple these two effects is to interface the active Mott channel with a charge transfer layer with intrinsically lower carrier density. The presence of such a charge transfer layer can modify the net field effect through the following mechanisms: (I) it reduces the net carrier density in the Mott channel that participates in the field effect modulation; (II) it creates a tailored carrier density profile in the Mott channel, with the high carrier density at the PZT/$R$NiO$_3$ interface (and the associated scaling advantage) preserved; (III) it serves as an extended screening layer that can mitigate depolarization; and (IV) it provides a parallel conduction channel as a shunting resistor, which can also attenuate the overall field effect. By optimizing the material choice and layer thickness of the bilayer channel, it is possible to enhance the first three effects and minimize the last one to achieve significantly enhanced resistance modulation in the Mott FeFETs.

To test this hypothesis, we compare the ferroelectric switching in single-layer and bilayer LaNiO$_3$ channels with the same $t_{\text{tot}}$. Figure 3b shows a series of resistance switching taken on a 4 uc LaNiO$_3$ upon applying voltage pulses with alternating signs, which corresponds to a $\Delta R/R_{\text{on}}$ of about 127%. In contrast, for a bilayer channel with comparable thickness, LNO(3)/LSMO(1), we observe a more than 80 times enhanced $\Delta R/R_{\text{on}}$ of 10,315% (Fig. 3c). This value is also one order of magnitude higher than that observed on the 3 uc single layer LaNiO$_3$ (930%, Supplementary Fig. 7b). The fact that inserting an additional

conduction layer enhances rather than attenuates the overall field effect demonstrates that mechanism IV is not a dominating factor, i.e., LSMO cannot be treated as an independent shunting resistor. We further examine devices with slightly higher $t_{tot}$ (Fig. 3d, e). If the bottom layer merely serves as a shunting resistor, increasing $t_{tot}$ would lead to reduced field effect modulation. Indeed, for single layer LaNiO$_3$, the 5 uc channel yields a $\Delta R/R_{on}$ of 30% (Fig. 3d), smaller than that in the 4 uc channel. For the bilayer channel, in sharp contrast, increasing LSMO thickness ($t_{LSMO}$) by 1 uc leads to a significantly higher field effect. Figure 3e shows the resistance switching hysteresis taken on a LNO(3)/LSMO(2) sample, where $\Delta R/R_{on}$ reaches 30,016%, clearly demonstrating the critical role of interfacial synergy between LaNiO$_3$ and LSMO in promoting the field effect modulation.

It is interesting to note that the off-state resistance of LNO(3)/LSMO(1) and LNO(3)/LSMO(2) samples settles at similar values, which is likely limited by the depolarization effect of the PZT gate. The on-state resistance of LNO(3)/LSMO(1) and LNO(3)/LSMO(2) samples is much higher than that of the single layer LNO(3) channel (Supplementary Fig. 7b). As ultrathin LSMO is too insulating ($R_\square \sim 530$ M$\Omega$ for 2 uc LSMO), it cannot provide sufficient parallel conduction to compensate for the reduced conduction in LaNiO$_3$ due to the charge transfer effect.

Figure 3f shows the time-dependence of the normalized on- and off-state resistance, defined as $R_{norm} = [R(t)-R_{on}]/[R_{off}-R_{on}]$, taken on a PZT/LNO(3)/LSMO(2) sample. Resistance in both polarization states shows an initial relaxation and saturates after about 200 seconds. Within the first 20 seconds, the off-state resistance rapidly drops to about 70% of the initial value. In contrast, the on-state resistance only increases by about 10%. This result clearly shows that the higher carrier density (i.e., on state) can directly correlate with a smaller resistance relaxation, consistent with the depolarization mechanism. The final resistance on/off ratio settles at about 55% of the initial value. The relaxation time and saturation level are comparable with previously reported values for Sm$_{0.5}$Nd$_{0.5}$NiO$_3$ single-layer channel[5] and Sm$_{0.5}$Nd$_{0.5}$NiO$_3$/LSMO bilayer channel[12] devices, despite the significantly higher $r_{off/on}$ observed in this sample.

Next, we examine the cycling behavior of the sample by applying ±5 V voltage pulses via a functional generator. As shown in Fig. 3g, the resistance on/off ratio is stable up to $10^6$ cycles and gradually decreases to about 76% of the initial value after $10^8$ cycles. It then remains at this level till $10^{10}$ cycles. This is consistent with the characteristic three-stage fatigue behavior, i.e., (I) slow fatigue stage, (II) logarithmic stage, and (III) saturated stage[37]. The endurance and saturation value, on the other hand, well surpass devices based on polycrystalline PZT films, supporting the scenario that the film-electrode interface plays a critical role in determining the fatigue behavior[37]. The robust retention and cycling behaviors of these Mott FeFETs make them highly competitive for nanoelectronic applications.

## Interfacial synergy between $R$NiO$_3$ and LSMO

To understand the relevant length scales of the interfacial interaction between $R$NiO$_3$ and LSMO, we examine the field effect in devices with various layer thicknesses. Figure 4a shows $\Delta R/R_{on}$ vs. $t_{LSMO}$ for bilayer channels with LaNiO$_3$ layer thickness ($t_{LNO}$) ranging from 3 to 5 uc. With the same $t_{LNO}$, the bilayer channels consistently exhibit higher $\Delta R/R_{on}$ than the single-layer channels ($t_{LSMO} = 0$), even when the LSMO layers are as thick as 5 uc. A similar trend is observed in the NdNiO$_3$ samples, where inserting a 2–5 uc LSMO buffer layer enhances the ferroelectric field effect in 4 and 5 uc NdNiO$_3$ channels (Fig. 4b). For 2 uc and thicker LSMO layers, such enhancement increases with decreasing $t_{LSMO}$, which is consistent with the interfacial nature of the screening effect. For the bilayer channels with 3 uc LaNiO$_3$, further reducing $t_{LSMO}$ to 1 uc leads to a lower $\Delta R/R_{on}$, confirming mechanism III, i.e., the composite channel reaches the thickness scaling limit imposed by the depolarization of the ferroelectric gate.

Figure 4c shows the $t_{RNO}$-dependence of the field effect. For $R$NiO$_3$ channels interfaced with 2 and 5 uc LSMO, $\Delta R/R_{on}$ increases with decreasing $t_{RNO}$ till the dead layer thickness, consistent with the field effect scaling behavior. For the LaNiO$_3$/LSMO(2) series, $\Delta R/R_{on}$ reaches the peak value of 38,440% in the 2.5 uc LaNiO$_3$ channel and drops slightly as $t_{LNO}$ is reduced to 2 uc, which is the electric dead layer thickness.

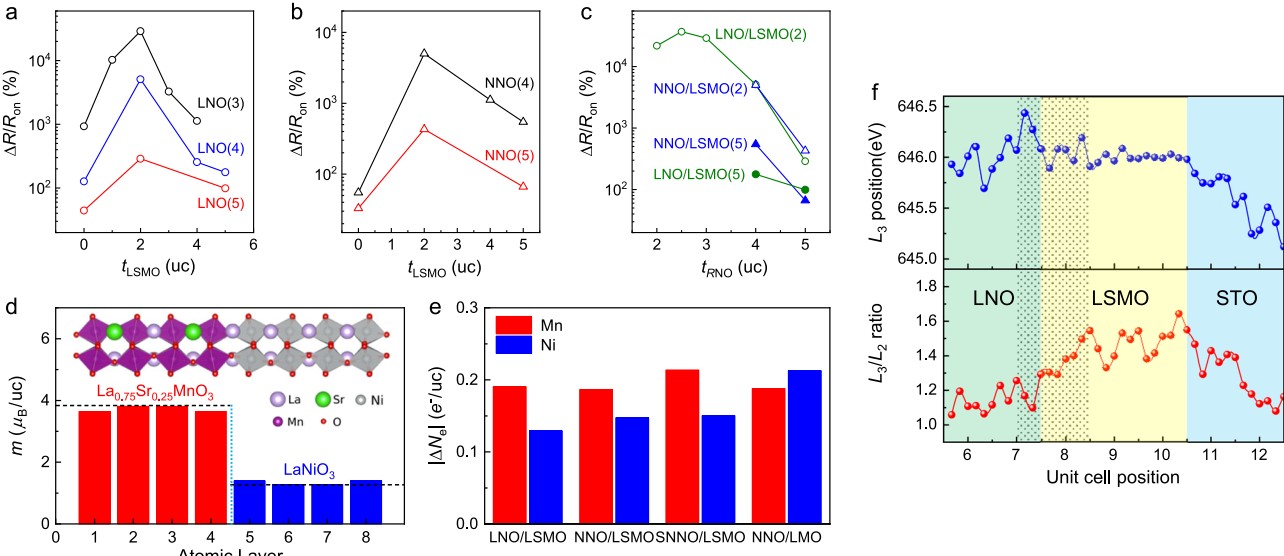

**Fig. 4 | Thickness dependence of conduction and field effect in $R$NiO$_3$/LSMO bilayer channels. a, b** $\Delta R/R_{on}$ vs. $t_{LSMO}$ for LaNiO$_3$/LSMO **a** and NdNiO$_3$/LSMO **b** with different $R$NiO$_3$ layer thicknesses. **c** $\Delta R/R_{on}$ vs. $t_{RNO}$ for LaNiO$_3$ and NdNiO$_3$ interfaced with 2 and 5 uc LSMO. **d** DFT + $U$ calculation of projected magnetic moments on Mn or Ni atoms in the unit of Bohr magneton ($\mu_B$) as a function of atomic layer number in a LaNiO$_3$(4)/La$_{0.75}$Sr$_{0.25}$MnO$_3$(4) superlattice. The dashed lines illustrate the average magnetic moments in bulk LaNiO$_3$ and La$_{0.75}$Sr$_{0.25}$MnO$_3$. Inset: Schematic of the supercell. **e** Calculated charge transfer values for $R$NiO$_3$(4)/L(S)MO(4) superlattice. The SNNO(4)/LSMO(4) data are taken from ref. 12. **f** Mn $L_3$ peak position (top) and $L_3/L_2$ intensity ratio (bottom) as a function of unit cell position taken on a PZT/LNO(3)/LSMO(3) sample. The data are extracted from the EELS Mn spectra shown in Supplementary Fig. 3. The shadowed area highlights the region with large charge transfer and serves as a guide to the eye.

Overall, the layer-thickness dependence of the field effect can be attributed to the interplay between two interfacial effects that tailor the carrier density profile in $R$NiO$_3$. At the PZT/$R$NiO$_3$ interface, the ferroelectric field effect modulates the carrier density in $R$NiO$_3$ within the screening length, leading to the exponential decay of the field effect with $t_{RNO}$[36]. At the $R$NiO$_3$/LSMO interface, another important length scale is the charge transfer length $L_{ct}$. In previous studies, the charge transfer effect has been intensively investigated in various nickelate/manganite heterointerfaces, including the LaNiO$_3$/La$_{2/3}$Sr$_{1/3}$MnO$_3$[38] and LaNiO$_3$/LaMnO$_3$ (LMO)[15,39,40] superlattices and Sm$_{0.5}$Nd$_{0.5}$NiO$_3$/La$_{0.67}$Mn$_{0.33}$MnO$_3$[12] and La$_{0.7}$Sr$_{0.3}$MnO$_3$/NdNiO$_3$[41] heterostructures. In these studies, x-ray absorption spectroscopy studies show a consistent blue shift of Mn $L_3$ edge by 0.7–1.15 eV[12,15,39], corresponding to a nominal valence change of Ni$^{3+}$ + Mn$^{3+}$→ Ni$^{2+}$ + Mn$^{4+}$ by about 0.06-0.1 electron/Mn. The charge coupling has led to emergent antiferromagnetic/ferromagnetic states[15,39,41], noncolinear magnetic structure[38], and interfacial exchange coupling[41]. The reported $L_{ct}$ varies from 1 to 4 monolayers[12,15,38–40]. The charge transfer can effectively reduce the net carrier density in the $R$NiO$_3$ channel by an amount comparable with the polarization field of PZT, leading to a substantial boost in the field effect. As the tailoring effect decays with distance from the $R$NiO$_3$/LSMO interface, it does not compromise the high carrier density at the PZT/$R$NiO$_3$ interface, thus retaining the size scaling advantage of the active Mott channel.

To understand the charge transfer effect quantitatively, we perform spin-polarized DFT + $U$ calculations of $R$NiO$_3$/La$_{0.75}$Sr$_{0.25}$MnO$_3$ superlattices ($R$ = La and Nd) composed of 4 uc nickelates and 4 uc manganites (insets of Fig. 4d and Supplementary Fig. 14a). We note that due to metal-oxygen hybridization, transfer of conduction electrons always accompanies redistribution of valence electrons, known as the rehybridization effect[42,43]. Therefore, one cannot directly use the change of the transition metal $d$-orbital occupancy to estimate the transfer of conduction electrons across the interface of $R$NiO$_3$(4)/La$_{1-x}$Sr$_x$MnO$_3$(4) ($R$ = La, Nd) heterostructures. Previous studies have shown that in a spin-polarized DFT calculation, the change of magnetic moment of the transition metal $d$-orbital can be used as a reasonable estimate of conduction electron transfer[12,44,45]. Figure 4d shows the layer projected magnetic moments in the LaNiO$_3$(4)/La$_{0.75}$Sr$_{0.25}$MnO$_3$(4) superlattice as well as the average magnetic moments in bulk LaNiO$_3$ and La$_{0.75}$Sr$_{0.25}$MnO$_3$ (dashed lines), which clearly illustrate the loss (gain) of conduction electrons in the interfacial Mn (Ni) layers. To account for the mixed termination of our samples, we average the magnetic moments of both interfaces. By comparing with the bulk values of LaNiO$_3$ and La$_{0.75}$Sr$_{0.25}$MnO$_3$, we deduce a conduction electron transfer $|\Delta N_e|$ of about 0.19 $e^-$ per Mn and 0.13 $e^-$ per Ni in the LaNiO$_3$(4)/La$_{0.75}$Sr$_{0.25}$MnO$_3$(4) superlattice (Fig. 4e). For the NdNiO$_3$(4)/La$_{0.75}$Sr$_{0.25}$MnO$_3$(4) superlattice, the $|\Delta N_e|$ values are about 0.19 $e^-$ per Mn and 0.15 $e^-$ per Ni. The loss of conduction electrons in Mn ions may not necessarily be equal to the gain of conduction electrons in Ni ions because the Mn-O and Ni-O hybridization can be different, and some electrons may migrate into the O−2$p$ orbitals during the charge transfer. These values are comparable to and slightly smaller than those obtained for the Sm$_{0.5}$Nd$_{0.5}$NiO$_3$(4)/LSMO(4) superlattice[12]. Reducing the Sr content, e.g., replacing LSMO with LaMnO$_3$ (LMO), can also lead to larger charge transfer (Supplementary Fig. 14b).

We further investigate the charge transfer effect in the PZT/LNO(3)/LSMO(3) sample by analyzing the EELS Mn $L$ spectra (Supplementary Fig. 3). The Mn-$L$ edges are observed within 2 uc distances below and above the LSMO interfaces. In addition to the intermixing across the interface (Mn diffusion) and atomic scale interface terraces, core-loss signal delocalization, electron probe dechanneling due to the atomic distortion near/at the interface, and multiple scattering can also significantly contribute to the broadening of the EELS signal[46]. As shown in Fig. 4f, we observe an enhanced blue shift of the $L_3$ peak position and suppressed $L_3/L_2$ intensity ratio towards the LaNiO$_3$/LSMO interface, which can be attributed to the formal valence change from Mn$^{3+}$ to Mn$^{4+}$. This result clearly reveals the electron transfer from Mn to Ni ions (or hole transfer from Ni to Mn). The change occurs predominantly within 1–2 unit cells at the interface with LaNiO$_3$, in good agreement with the theoretical prediction (Fig. 4e). The $L_3$ peak position shifts by about 0.7 eV, which corresponds to about 0.06 electron/Mn. This value estimates the lower bound of the charge transfer effect, as the $L_3$ position for the entire film thickness of LSMO (3 uc) may be modulated compared with the standalone LSMO film. For example, the field effect results suggest that the charge transfer effect persists in bilayer channels with up to 5 uc LSMO (Fig. 4a-c and ref. 12). Both the charge transfer amount and the length scale of valence distribution are in good agreement with previous reports[12,15,38–40].

In Fig. 4e, the weak dependence of the charge transfer on the $A$-site cation explains the comparable field effect observed in the 4 and 5 uc LaNiO$_3$ and NdNiO$_3$ samples interfaced with 2 uc LSMO (Fig. 4c). On the other hand, the bilayer channels with LaNiO$_3$ can be further scaled down as its electric dead layer thickness is thinner than that of NdNiO$_3$ and thus can lead to higher field effect. Above the dead layer thickness, the enhancement of the field effect is most effective when there is a sizable overlap between the field effect screening region and the charge transfer region, which sets the optimal length for $t_{RNO}$. As LSMO compensates for the reduced screening capacity of the $R$NiO$_3$ layer due to the reduced carrier density, the optimal LSMO thickness for the field effect depends on the competing requirements of minimizing depolarization field, which calls for larger $t_{LSMO}$, and minimizing parallel conduction. The conduction in LSMO can be divided into two regions. For the top layer interfaced with $R$NiO$_3$, the charge transfer effect would suppress rather than enhance conductivity, as $x$ = 0.33 is close to the optimal doping. Increasing $x$ by 0.1/uc tunes LSMO close to the half-doping insulating phase[2]. This means the LSMO layer within the charge transfer length $L_{ct}$ does not effectively participate in the parallel conduction. As the bottom LSMO layer beyond $L_{ct}$ is unaffected by the ferroelectric field effect and only weakly affected by the charge transfer effect, it can be viewed as an independent shunting resistor. From Fig. 4a, b, $\Delta R/R_{on}$ peaks at $t_{LSMO}$ = 2 uc, suggesting the $L_{ct}$ is on the order of 2 uc, which is also consistent with the theoretical calculation (Fig. 4e) and EELS results (Fig. 4f).

## Comparison of ferroelectric field effect in various correlated oxides

Figure 5a summarizes the $\Delta R/R_{on}$ results taken on $R$NiO$_3$/LSMO bilayer channels with different layer thicknesses. Despite the difference in the $A$-site element and $t_{RNO}/t_{LSMO}$ combination, $\Delta R/R_{on}$ follows strikingly similar exponential scaling with the channel thickness $t_{tot}$. Such a universal scaling behavior can serve as a guideline for designing complex oxide-based Mott FeFETs. Figure 5b compares the maximum $\Delta R/R_{on}$ vs. channel thickness $t_{channel}$ obtained on ferroelectric-gated various correlated oxide systems, including $R$NiO$_3$/LSMO bilayers[12], single-layer nickelates, ruthenate[47], manganites[14,24], cobaltate[48], and cuprates[19,49]. The largest room temperature resistance on-off ratio $r_{off/on}$ = 385.4 is observed in the LaNiO$_3$(2.5)/LSMO(2) bilayer channel (Fig. 5b inset), more than 30 times higher than the best result for Mott FeFET reported in the literature, which is obtained on termination-controlled single-layer LaNiO$_3$ devices[7]. For the NdNiO$_3$/LSMO heterostructure, the largest $r_{off/on}$ is 54.2 obtained on the NNO(4)/LSMO(2) sample (Supplementary Fig. 13a). For both systems, the device performance shows significant advantages over conventional single-channel Mott FETs in terms of on-off ratio, size scaling, and retention[5]. It is also highly competitive compared to the state-of-the-art MRAM technology[50], promising for developing voltage-controlled nonvolatile memory for low-power operation.

In summary, we have systematically studied the ferroelectric field effect in rare earth nickelate-based Mott channels. By inserting a 1–5

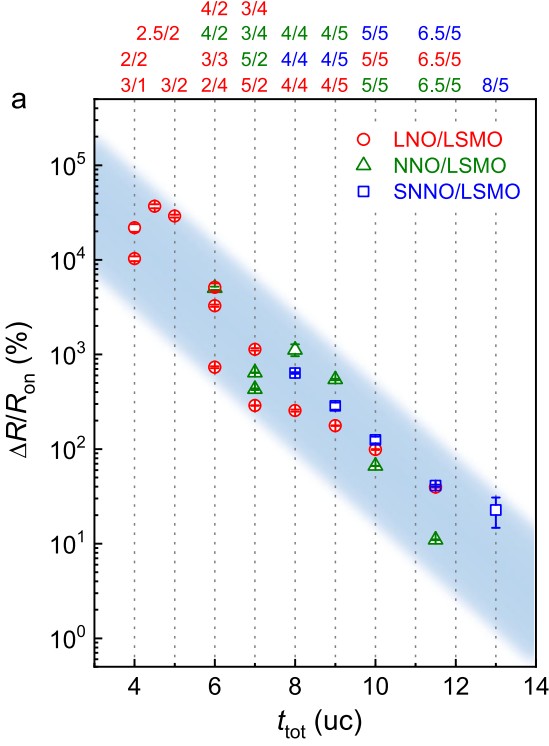

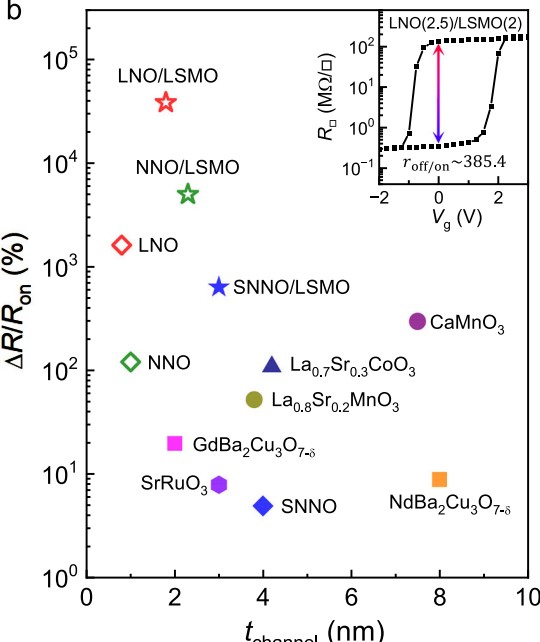

**Fig. 5 | Room temperature ferroelectric field effect in various correlated oxide channels. a** $\Delta R/R_{on}$ vs. $t_{tot}$ in $R$NO/LSMO channels (error bars: SD), with the layer thickness combination labelled above the plot. **b** Maximum $\Delta R/R_{on}$ vs. $t_{channel}$ achieved in various correlated oxide materials, including $R$NO/LSMO bilayers[12] (stars), single-layer nickelates (diamonds), ruthenate[47] (hexagon), manganites[14,24] (circles), cobaltate[48] (triangle), and cuprates[19,49] (squares). The open symbols are reported in this work. Inset: $R_{\square}$ vs. $V_g$ switching hysteresis taken on the PZT/LNO(2.5)/LSMO(2) sample.

unit cell thick charge transfer layer, we have achieved record high room temperature resistance modulation in Mott FeFETs. These devices show superior retention and cycling behavior. The performance depends on the collective effects of optimizing charge screening in the correlated channel and minimizing depolarization for the ferroelectric gate. Leveraging the highly competitive device parameters, scalable epitaxial growth, voltage-controlled nature, and potential for integrating complex oxides with the Si platform, our device scheme offers an effective material strategy for developing next-generation high-density, low-power nanoelectronics and spintronics.

## Methods
### Sample preparation and device fabrication
We deposited epitaxial perovskite oxide thin films and heterostructures via off-axis radio frequency magnetron sputtering. The LSMO layer was grown at 650 °C in 150 mTorr gas (Ar: $O_2$ = 2:1). The $LaNiO_3$, $NdNiO_3$, $SmNiO_3$, and PZT layers were deposited at 500 °C with process gas of 60 mTorr (Ar: $O_2$ = 1:2), 80 mTorr (Ar: $O_2$ = 1:2), 135 mTorr (Ar: $O_2$ = 1:2), and 120 mTorr (Ar: $O_2$ = 2:1), respectively. We pre-patterned the substrates into the Hall bar device geometry via photolithography and then deposited 15 nm Ti. The channel size varies from $80 \times 40$ μm² to $10 \times 5$ μm². After lift-off, the substrates were annealed at 400 °C for 4 hours with the Ti layer fully oxidized to $TiO_x$. After in situ deposition of the heterostructure, a 30 nm Au layer was deposited as electrodes. The detailed device fabrication process is discussed in Supplementary Note 2.

### Sample characterization
The transport studies were carried out in a Quantum Design Physical Property Measurement System, with the channel resistance characterized in four-point geometry via Keithley 2400 SourceMeter or Stanford Research SR830 lock-in at current ≤20 μA. The AFM and PFM studies were carried out using a Bruker Multimode 8 AFM. Conductive AFM and PFM measurements were conducted in contact mode (tip: SCM-PIT cantilever with Platinum-Iridium coating). Ferroelectric domains were written with a bias voltage ($V_{bias}$) of ±8 V with respect to the bottom electrode. The PFM imaging was taken with 600 mV AC voltage at around 300 kHz, close to the tip resonance frequency.

### STEM and EELS studies
The TEM samples were prepared using the focused ion beam (FIB) with 2 keV Ga+ ion as a final milling. The STEM and EELS were performed using a JEOL ARM 200CF microscope equipped with both image and probe correctors. The elemental mapping was performed using the direct electron detector K3 (Gatan, Inc.) to simultaneously detect a large range of elemental energy losses, including Pb-$M_{4,5}$ edges ~2500 eV, with 0.9 eV/channel energy dispersion. For the Mn valence state analysis, the energy resolution was ~0.7 eV based on the measured full width at half maximum (FWHM) of the zero-loss peak with 0.09 eV/channel. The acquired EELS data were denoised by the Principal Component Analysis in the Digital Micrograph (Gatan, Inc.). The collection angles for high-angle annular dark-field STEM images were in the range of 68–280 mrad.

### First-principles DFT calculations
We performed density functional theory (DFT) calculations using Vienna ab initio Simulation Package (VASP)[51,52] with a plane-wave basis set and projector augmented wave pseudopotentials[53]. We used generalized gradient approximation with the Perdew–Burke–Ernzerhof (PBE) parameterization for the exchange-functional and an energy cutoff of 600 eV. To simulate the $LaNiO_3(4)/La_{0.75}Sr_{0.25}MnO_3(4)$ and $NdNiO_3(4)/La_{1-x}Sr_xMnO_3(4)$ superlattices, we used $\sqrt{2} \times \sqrt{2} \times 8$ supercells (see Fig. 4 and Supplementary Note 8) to accommodate the rotation and tilt of oxygen octahedra. The in-plane cell lattice constant of those superlattices were fixed to be the theoretical lattice constant

of SrTiO$_3$ (3.94 Å) in order to simulate the epitaxial growth on SrTiO$_3$ substrate. The out-of-plane cell lattice constant and internal coordinates were fully relaxed until each force component was less than 10 meV/Å and the stress tensor was smaller than 10 kBar. We used a Monkhorst–Pack grid of $8 \times 8 \times 1$ to sample the Brillouin zone of the superlattice supercells. To study bulk La$_{1-x}$Sr$_x$MnO$_3$, LaNiO$_3$, and NdNiO$_3$, we used a 20-atom simulation cell and a Monkhorst–Pack grid of $8 \times 8 \times 6$ for the Brillouin zone integration. We adopted the rotationally invariant DFT + $U$ method[54] to take into account the correlation effects on $3d$ orbitals of transition metal ions. Specifically, we used $U = 5.0$ eV and $J = 0.7$ eV for both Mn and Ni $3d$ orbitals. We also added $U = 9.0$ eV and $J = 0.0$ eV on the La $4f$ orbitals to shift the empty $4f$ states to higher energy. These $U$ and $J$ are reasonable values for first-row transition metal ions and are used in previous works[12]. As one cannot directly use the change of transition metal $d$-orbital occupancy to estimate the transfer of conduction electrons between two correlated oxides[36,55], we used the change of magnetic moment of the transition metal $d$-orbital to estimate the conduction electron transfer in a spin-polarized DFT calculation[12,44,45] and obtained a ferromagnetic state in $R$NiO$_3$(4)/La$_{0.75}$Sr$_{0.25}$MnO$_3$(4) (or LaMnO$_3$) ($R$ = La, Nd) superlattices throughout the calculations[12,44,45].

## Data availability

The data generated in this study are presented in the main text and Supplementary Information. The raw data that support the plots within this paper and other findings of this study are available from the corresponding author upon reasonable request.

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

## Acknowledgements

We would like to thank Kun Wang and Alex Pofelski for technical assistance. This work was primarily supported by the National Science Foundation (NSF) Grant No. DMR-1710461 and EPSCoR RII Track-1: Emergent Quantum Materials and Technologies (EQUATE), Award No. OIA–2044049, and Semiconductor Research Corporation (SRC) under GRC Task Number 2831.001. H.C. is supported by the National Key R&D Program of China (2021YFE0107900) and Science and Technology Commission of Shanghai Municipality under grant number 23ZR1445400. Y.-W.F. and H.C. acknowledge the support of computational resources provided by the HPC centers of NYU Shanghai and NYU New York. The electron microscopy work at the Brookhaven National Laboratory was supported by Division of Materials Science and Engineering, Office of Basic Energy Science, U.S. Department of Energy under Contract No. DE-SC0012704. TEM sample preparation was performed at the Center for Functional Nanomaterials, Brookhaven National Laboratory. The research was performed in part in the Nebraska Nanoscale Facility: National Nanotechnology Coordinated Infrastructure and the Nebraska Center for Materials and Nanoscience, which are supported by NSF under Award ECCS: 2025298, and the Nebraska Research Initiative.

## Author contributions

X.H. conceived and supervised the project. Y.H., X.C., and L.Z. prepared the oxide heterostructures. Y.H. and X.C. performed sample characterization, device fabrication, and electrical measurements. Y.-W.F. and H.C. performed the DFT modeling. M.-G.H., W.W., and Y.Z. performed the TEM studies. Y.H. and X.H. wrote the manuscript. All authors discussed the results and contributed to the manuscript preparation.

## Competing interests

The authors declare no competing interests.
