## [Peer Review File · Nature Communications]

REVIEWER COMMENTS

Reviewer #1 (Remarks to the Author):

In the manuscript entitled “Harnessing the competing roles of charge to design ferroelectric-gated Mott transistors” by Y. Hao et al., the authors demonstrate large, room-temperature resistance switching in ferroelectric field effect Mott transistors. The room-temperature R_{off}/R_{on} ratios of more than 300 presented here represent a clear breakthrough in the field, which for much of the last 20 years had been stuck with single digit on-off ratios (and only recently demonstrated on/off ratios above 10). In addition, this work suggests clear research directions to continue optimizing the channel materials to tailor the charge profile within the device, and potentially achieve even larger room-temperature switching. The manuscript is clearly written, the samples are of high quality, the results are clearly presented and well-analyzed, and oxide heterostructure growth and characterization remains a highly-active field of condensed matter research. For these reasons, the work is likely to be of interest to the readership of Nature Communications, and I support publication with only a few minor comments that the authors should consider addressing.

- In Fig. 3a & b (and similar figures in the SI), the resistance decays immediately after switching. Is this due to the effects of the depolarization field? And, if so, do you expect the resistance decay to be different between the on and off states if they have very different carrier concentrations?
- It has previously been shown that depositing a capping layer on nickelate thin films can suppress structural distortions at the surface, and enhance the conductivity [D. Kumah et al. *Adv. Mater.* 26 1935 (2014)]. Could a similar mechanism be important here when demonstrating that the largest switching in single layer $RNiO_3$ channels occurs for films with thickness comparable to the dead layer thickness (Fig. 2e) determined from uncapped films (Figs. 2a–c).
- While the authors cite M. Gibert et al. (Ref. 15), there have been several additional studies that looked at charge transfer at the nickelate / manganite interface and attempted to quantify both the number of charges involved, relevant length scales, and consequences. It may be useful to compare the values derived from DFT calculations with experimentally measured values of similar systems.

In summary, I fully support publication of the manuscript in Nature Communications after the authors consider the minor comments and questions posed above.

Reviewer #2 (Remarks to the Author):

Hao et al report on ferroelectric field effect devices combining PZT with a correlated oxide channel based on nickelate single layers or manganite/nickelate bilayers. The key result is a giant enhancement of the resistance contrast in bilayer systems, which is an important result for the community and worth of publication in Nature Communications. However, the mechanism behind the resistive switching is not fully clarified. Please find detailed comments below.

1. The paper title does not highlight the main result of the paper which is the realization of a record high resistance contrast in correlated oxide FeFETs.
2. The first sentence of the abstract is too long and the formulation « while the close to metallic carrier density of the strongly correlated channel precludes a substantial field effect modulation via a solid-state gate » looks weird.
3. The paper does not read very well and I strongly suggest the authors to have it proofread by a native speaker (or use AI tools to improve it).
4. Fig S2. What is the layer on top of the PZT ? Where are the LNO and LSMO on panel a ?
5. The lithography process is not explained with sufficient detail. What is the role of the Ti layer ? Does it influence the PZT ? The authors mention a lift-off process. What is lifted off ? How ? Please include sketches of the lithography workflow in the supplementary material, with all technical details so that the work can be reproduced by others in the community.
6. What is the device size ? How does the effect scale upon reducing it ?
7. What is the substrate for the films in Fig 2 ?
8. The discussion on the expected modulation amplitude of the carrier density by the PZT is interesting but what is critically missing in this otherwise very nice paper is measurements of the carrier density through the Hall effect, in single films and in devices for the up and down polarization states. This information would bring key insight into the mechanism responsible for the giant resistance switching. The authors are requested to perform such measurements.
9. Similarly the authors discuss the charge transfer at the LSMO/nickelate interface from literature results. It would be very interesting to measure the Ni and Mn valence through EELS in the STEM specimens. Or alternatively to perform XPS or XAS. This information would also strengthen the author's claims on the underlying mechanisms (and confirm the DFT results).
10. More quantitative information on the retention time and fatigue, especially for the best devices, in comparison with the literature.

In summary, this paper could be published in Nature Comms after major revisions and the inclusion of Hall and spectroscopy data to ascertain the mechanism at play in the proposed devices.

Reviewer #3 (Remarks to the Author):

Referee report on manuscript NCOMMS-23-23990-T

In this paper, the authors study ferroelectric-gated Mott transistors based on ferroelectric PZT and nickelate / manganite correlated oxides.

Ferroelectric Mott transistors display non-volatile switching and potentially large changes in resistance when switching due to the large (localized) carrier density present in the insulating Mott state. Here the authors report on an extremely large change in resistance when switching the ferroelectric polarization (up to a factor of about 400) in a device combining, for the channel, LaNiO₃ (and NdNiO₃, SmNiO₃) – a metallic high carrier density material – and LaSrMnO₃ a material known for its colossal magnetoresistance.

This is a detailed complete study on very many samples exploring the role of the thickness of the nickelate compounds and the one of the LaSrMnO₃. The record high resistance switching is obtained when the nickelate layer thickness is close to the thickness at which the material displays a “dead layer” (insulating) behavior – 2.5 unit cells and a LaSrMnO₃ thickness of 2 unit cells.

This is nice paper describing an interesting detailed study of Mott-like transistors using complex oxide heterostructures. The addition of a third LaSrMnO₃ layer in the heterostructure leads to record high resistance switching. The paper can be considered for publication in Nature communications. I have however several questions / comments that the authors should address.

Comments / questions:

-I start with an important comment / question on the physics of the studied devices. No doubt that the results are spectacular but there are a few things I do not understand. A Mott transistor is a device in which the channel is in a Mott insulating state (say $1e^-$ per unit cell). The carrier density is extremely high as stated by the authors and the interest is that, under the gate applied electric field, one is moving away from half filling (the Mott state) producing a deconfinement and then conduction takes place with all the electrons that were localized (not only the ones coming from the field

effect). So one can restore conduction with a change in carrier density much smaller than the carrier density of the Mott state – unlike what is stated p. 3.

Noticing that NdNiO₃ and SmNiO₃ are thought to be size selective Mott insulators (below about 400K for SmNiO₃ and 200K for NdNiO₃) and LaNiO₃ a paramagnetic metal – not a Mott insulator. My questions are:

Regarding the mechanism proposed and the role of the LaSrMnO₃, the authors say that a charge transfer occurs reducing the carrier density of the ReNiO₃ layer. The idea is, as I understand, that this lowering of the carrier density reduces the sheet carrier density boosting the ferroelectric field effect without producing a parallel conduction (low conductivity in LaSrMnO₃).

I can understand the reasoning but not the link to a Mott transistor. If a charge transfer occurs, reducing the ReNiO₃ charge density, the system should move away from the Mott state and thus restore conduction (the mechanism of the Mott transistor). With LaSrMnO₃, are the authors claiming that the NdNiO₃ (which is not in a Mott state at RT) and SmNiO₃ layers are in a Mott insulating state at RT?

LaNiO₃ is not a Mott insulator – it is metallic compound. Are the authors suggesting that ultrathin LaNiO₃ layers, at the verge of the insulating state, are in a Mott insulating state?

Other comments:

-I do not understand the comment bottom of p.3 : “... how to satisfy the requirement of reducing the effective sheet carrier density while preserving screening?” – this requirement is in principle satisfied with a Mott transistor, the change in carrier density to produce conduction being much smaller than the total carrier density as already mentioned above.

-LaNiO₃ is rhombohedral while, at room temperature, NdNiO₃ and SmNiO₃ are orthorhombic, this might be worth mentioning.

-The authors mention a very high polarization for PZT, 80microC/cm². Was this measured? What is the estimate of the polarization obtained from the resistance switching (obtained from the change in sheet carrier density)?

-Have the authors characterize the behavior of single layers of LaSrMnO₃? What is the sheet resistance of the LaSrMnO₃ layer?

-P.4 a detail – channel not channel.

Manuscript NCOMMS-23-23990-T: Response to Reviewers' Comments

We thank the reviewers for reviewing our manuscript and appreciate their critical comments and valuable suggestions. All three reviewers made very positive comments on the scientific quality and potential impact of our work. Reviewer #1 stated that *“The room-temperature R_{off}/R_{on} ratios of more than 300 presented here represent a clear breakthrough in the field, which for much of the last 20 years had been stuck with single digit on-off ratios (and only recently demonstrated on/off ratios above 10). In addition, this work suggests clear research directions to continue optimizing the channel materials to tailor the charge profile within the device, and potentially achieve even larger room-temperature switching.”* Reviewer #2 commented that our key result of *“a giant enhancement of the resistance contrast in bilayer systems”* is *“an important result for the community and worth of publication in Nature Communications.”* Reviewer #3 commended that our approach *“lead to record high resistance switching”* and *“No doubt that the results are spectacular.”* The reviewers also raised questions about the conductivity and carrier density of the channel materials, the magnitude of ferroelectric polarization, the device retention and fatigue characteristics, the underlying physics of Mott transistors, and supporting evidence of the charge transfer effect. To address these comments, we fabricated new samples, performed additional electrical and EELS studies, and made substantial revisions to the main text and Supplementary Information (SI). Below is a point-by-point reply to the comments. A summary of the changes is listed at the end of the response letter. These changes are also highlighted in the main text and SI.

Response to Reviewer #1's Comments

“In the manuscript entitled “Harnessing the competing roles of charge to design ferroelectric-gated Mott transistors” by Y. Hao et al., the authors demonstrate large, room-temperature resistance switching in ferroelectric field effect Mott transistors. The room-temperature R_{off}/R_{on} ratios of more than 300 presented here represent a clear breakthrough in the field, which for much of the last 20 years had been stuck with single digit on-off ratios (and only recently demonstrated on/off ratios above 10). In addition, this work suggests clear research directions to continue optimizing the channel materials to tailor the charge profile within the device, and potentially achieve even larger room-temperature switching. The manuscript is clearly written, the samples are of high quality, the results are clearly presented and well-analyzed, and oxide heterostructure growth and characterization remains a highly-active field of condensed matter research. For these reasons, the work is likely to be of interest to the readership of Nature Communications, and I support publication with only a few minor comments that the authors should consider addressing.”

Comment 1.1:

“1. In Fig. 3a & b (and similar figures in the SI), the resistance decays immediately after switching. Is this due to the effects of the depolarization field? And, if so, do you expect the resistance decay to be different between the on and off states if they have very different carrier concentrations?”

Reply 1.1:

We thank the reviewer for the positive comments and valuable suggestions, which help us improve the technical strength and clarity of the paper. We do believe the resistance decay immediately after polarization switching is due to depolarization. Figure R1.1 shows the time-dependence of the normalized on- and off-state resistance taken on a PZT/LaNiO₃ (3 uc)/La_{0.67}Sr_{0.33}MnO₃ (2 uc) sample. Resistance in both polarization states shows an initial relaxation and saturates after about 200 seconds. Within the first 20 seconds, the off-state resistance rapidly drops to about 70% of the initial value. In contrast, the on-state

resistance only increases by about 10%. This result clear shows that the higher carrier density (*i.e.*, on state) can be directly correlated with a lower resistance relaxation.

To clarify this point, we included Fig. R1.1 as Fig. 3f in the main text and added the following discussion to the 3rd paragraph on Page 10:

“Figure 3f shows the time-dependence of the normalized on- and off-state resistance, defined as $R_{\text{norm}} = [R(t) - R_{\text{on}}] / [R_{\text{off}} - R_{\text{on}}]$, taken on a PZT/LNO(3)/LSMO(2) sample. Resistance in both polarization states shows an initial relaxation and saturates after about 200 seconds. Within the first 20 seconds, the off-state resistance rapidly drops to about 70% of the initial value. In contrast, the on-state resistance only increases by about 10%. This result clear shows that the higher carrier density (*i.e.*, on state) can be directly correlated with a lower resistance relaxation, consistent with the depolarization mechanism.”

Fig. R1.1 | Time-dependence of the normalized on- and off-state resistance taken on a PZT/LNO(3)/LSMO(2) sample on STO. The initial on-state (defined as zero) value is not shown in the logarithmic scale. This figure is included as Fig. 3f in the main text.

Comment 1.2:

“2. It has previously been shown that depositing a capping layer on nickelate thin films can suppress structural distortions at the surface, and enhance the conductivity [D. Kumah et al. *Adv. Mater.* 26 1935 (2014)]. Could a similar mechanism be important here when demonstrating that the largest switching in single layer $R\text{NiO}_3$ channels occurs for films with thickness comparable to the dead layer thickness (Fig. 2e) determined from uncapped films (Figs. 2a–c).”

Reply 1.2:

We thank the reviewer for raising this important point. We also note that the on-state R_{\square} of PZT-gated single layer $R\text{NiO}_3$ channels is always higher than that of the uncapped film with the same thickness. For the PZT-gated 2 uc single layer LaNiO_3 channel (Supplementary Fig. 3a), the 300 K sheet resistance R_{\square} is $5.2 \times 10^5 \Omega/\square$ for the off state and $3.0 \times 10^4 \Omega/\square$ for the on state. For the uncapped film (Fig. 2a), R_{\square} at 300 K is $1.7 \times 10^4 \Omega/\square$, suggesting that the PZT top-layer suppresses the channel conductivity. Similar suppression in conduction has also been observed in the NdNiO_3 and SmNdO_3 channels. We believe, however, such a change shares a similar mechanism as that observed in *Adv. Mater.* 26 1935 (2014) (cited as Ref. 35), which uses an LaAlO_3 capping layer. As LaAlO_3 is closely lattice matched with $R\text{NiO}_3$, it can suppress the lattice distortion induced by surface reconstruction in uncapped films and hence enhance conductivity. The PZT

capping layer, in contrast, imposes substantial tensile strain. Previous studies have shown that the conductivity of $RNiO_3$ is substantially suppressed when subjected tensile strain (Refs. 29, 30). To clarify this point, we included the following discussion in the 3rd paragraph on Page 7:

“We also note that depositing the PZT top-layer suppresses the channel conductivity. The on-state R_{\square} of PZT-gated single layer $RNiO_3$ channels is always higher than that of the uncapped film with the same thickness. Similar suppression in conduction has also been observed in the $NdNiO_3$ and $SmNdO_3$ channels. Such a change may share a similar mechanism as that observed in Ref. [35], where a closely lattice matched capping layer $LaAlO_3$ can suppress the lattice distortion induced by surface reconstruction and hence enhance $LaNiO_3$ conductivity. The PZT capping layer, in contrast, imposes a large tensile strain, which suppresses the $RNiO_3$ conduction^{29,30}.”

Comment 1.3:

“3. While the authors cite M. Gibert et al. (Ref. 15), there have been several additional studies that looked at charge transfer at the nickelate/manganite interface and attempted to quantify both the number of charges involved, relevant length scales, and consequences. It may be useful to compare the values derived from DFT calculations with experimentally measured values of similar systems.”

Reply 1.3:

As the reviewer suggested, the charge transfer effect has been intensively examined in various nickelate/manganite heterointerfaces, including the $LaNiO_3/La_{2/3}Sr_{1/3}MnO_3$ (Hoffman *et al.*, *PRX* **6**, 041038 (2016), cited as Ref. 39) and $LaNiO_3/LaMnO_3$ (Refs. 15,41; Hoffman *et al.*, *PRB* **88**, 144411 (2013), cited as Ref. 40) superlattices and $Sm_{0.5}Nd_{0.5}NiO_3/La_{0.67}Mn_{0.33}MnO_3$ (Ref. 12) and $La_{0.7}Sr_{0.3}MnO_3/NdNiO_3$ (Chen *et al.*, *PR Mater.* **4**, 054408(2020), cited as Ref. 42) heterostructures. The charge coupling has led to emergent antiferromagnetic/ferromagnetic states (Ref. 15, 40, 42), noncolinear magnetic structure (Ref. 39), and interfacial exchange coupling (Ref. 42). In these studies, x-ray adsorption spectroscopy spectra show a consistent blue shift of Mn L_3 edge by 0.7-1.15 eV (Refs. 12, 15, 40, 41), corresponding to electron transfer from Mn to Ni ions (or hole transfer from Ni to Mn) by about 0.06-0.1 electron/Mn. The reported charge transfer length varies from 1-4 monolayers (Ref. 12, 15, 39-41). Both the amount of charge transfer and its length scale are in good agreement with our theoretical predictions. To clarify this point, we added new references and included the following discussions in the main text:

Page 12, third paragraph: “In previous studies, the charge transfer effect has been intensively investigated in various nickelate/manganite heterointerfaces, including the $LaNiO_3/La_{2/3}Sr_{1/3}MnO_3$ ³⁹ and $LaNiO_3/LaMnO_3$ (LMO)^{15, 40, 41} superlattices and $Sm_{0.5}Nd_{0.5}NiO_3/La_{0.67}Mn_{0.33}MnO_3$ ¹² and $La_{0.7}Sr_{0.3}MnO_3/NdNiO_3$ ⁴² heterostructures. In these studies, x-ray adsorption spectroscopy studies show a consistent blue shift of Mn L_3 edge by 0.7-1.15 eV^{12, 15, 40, 41}, corresponding to a nominal valence change of $Ni^{3+} + Mn^{3+} \rightarrow Ni^{2+} + Mn^{4+}$ by about 0.06-0.1 electron/Mn. The charge coupling has led to emergent antiferromagnetic/ferromagnetic states^{15, 40, 42}, noncolinear magnetic structure³⁹, and interfacial exchange coupling⁴². The reported L_{ct} varies from 1-4 monolayers^{12, 15, 39-41}.”

Page 14, second paragraph: “Both the charge transfer amount and the length scale of valence distribution are in good agreement with previous reports^{12, 15, 39-41}.”

Response to Reviewer 2's Comments

“Hao et al report on ferroelectric field effect devices combining PZT with a correlated oxide channel based on nickelate single layers or manganite/nickelate bilayers. The key result is a giant enhancement of the resistance contrast in bilayer systems, which is an important result for the community and worth of publication in Nature Communications. However, the mechanism behind the resistive switching is not fully clarified. Please find detailed comments below.”

Comment 2.1:

“1. The paper title does not highlight the main result of the paper which is the realization of a record high resistance contrast in correlated oxide FeFETs.”

Reply 2.1:

We thank the reviewer for the positive comments and valuable suggestions, which help us improve the technical strength and clarity of the paper. Following the reviewer's suggestion, we changed the title to:

“Record High Room Temperature Resistance Switching in Ferroelectric-Gated Mott Transistors Unlocked by Interfacial Charge Engineering”.

Comment 2.2:

“2. The first sentence of the abstract is too long and the formulation « while the close to metallic carrier density of the strongly correlated channel precludes a substantial field effect modulation via a solid-state gate » looks weird.”

Reply 2.2:

We changed this sentence as follows:

“The superior size and power scaling potential of ferroelectric-gated Mott transistors makes them promising building blocks for developing energy-efficient memory and logic applications in the post-Moore's Law era. The close to metallic carrier density in the Mott channel, however, imposes the bottleneck for achieving substantial field effect modulation via a solid-state gate.”

Comment 2.3:

“3. The paper does not read very well and I strongly suggest the authors to have it proofread by a native speaker (or use AI tools to improve it).”

Reply 2.3:

We thank the reviewer for raising this important point. We have thoroughly revised the manuscript to improve the presentation.

Comment 2.4:

“4. Fig S2. What is the layer on top of the PZT? Where are the LNO and LSMO on panel a?”

Reply 2.4:

We deposited Pt/Au on top of PZT for the focused ion beam sample preparation. In Fig. S2a, the 3 unit cell (uc) LaNiO₃ and 3 uc La_{0.67}Sr_{0.33}MnO₃ layers are sandwiched between PbZr_{0.2}Ti_{0.8}O₃ and the substrate.

Due to the ultrathin layer thickness, they are relatively invisible on this scale with a shadowed grey color. We added this information to Supplementary Fig. S2 and its caption.

Comment 2.5:

“5. The lithography process is not explained with sufficient detail. What is the role of the Ti layer? Does it influence the PZT? The authors mention a lift-off process. What is lifted off? How? Please include sketches of the lithography workflow in the supplementary material, with all technical details so that the work can be reproduced by others in the community.”

Reply 2.5:

Figure R2.1 shows the process flow for the fabrication of FET devices. We first define Hall bar devices with different channel dimensions using photolithography (Fig. R2.1a) and deposit about 15-20 nm Ti on top of the entire substrate (Fig. R2.1b). The sample is then sonicated in acetone for a couple of seconds followed by IPA rinsing to remove photoresist. The Ti layer deposited on the photoresist is removed during this process, known as lift-off. The patterned substrate is annealed at 400 °C in a muffle oven for 4 hours, with the Ti layer oxidized into amorphous TiO_x and becoming transparent (Fig. R2.1c). As the substrate is only exposed in the Hall bar device area, with the rest area covered by amorphous TiO_x, epitaxial oxide thin films can only form in the Hall bar area. After *in situ* deposition of ultrathin correlated oxide layers and thick PZT films on the patterned substrate (Fig. R2.1d), we perform the second photolithography to define electrodes for the Hall bar device and top gate (Fig. R2.1e). Scratches are made in the Hall bar electrode area to ensure connection of the Au contact to the correlated channel. After Au deposition, we perform a second lift-off to remove the photoresist as well as the Au layer on top (Fig. R2.1f). We now included the detailed device fabrication process and Fig. R2.1 in the Supplementary Note 2.

Fig. R2.1 | Schematic device fabrication flow. This figure is included in the SI as Supplementary Fig. 5.

Comment 2.6:

“6. What is the device size? How does the effect scale upon reducing it?”

Reply 2.6:

The device channel size varies from 80x40 μm² to 10x5 μm². We now include this information in the Methods section. We did not observe a noticeable dependence of resistance on/off ratio on the device

channel size. As we work with photolithography, $10 \times 5 \mu\text{m}^2$ is the smallest device dimension we can work with for high-precision resistance measurements, where the width of the Hall probes is reasonably smaller than the channel width and the separation between the Hall probes.

Comment 2.7:

“7. What is the substrate for the films in Fig 2?”

Reply 2.7:

In Fig. 2, we deposited single-layer $R\text{NiO}_3$ on the LaAlO_3 substrates. We now added it to the figure caption.

Comment 2.8:

“8. The discussion on the expected modulation amplitude of the carrier density by the PZT is interesting but what is critically missing in this otherwise very nice paper is measurements of the carrier density through the Hall effect, in single films and it devices for the up and down polarization states. This information would bring key insight into the mechanism responsible for the giant resistance switching. The authors are requested to perform such measurements.”

Reply 2.8:

Thanks for raising this important point. Following the reviewer’s suggestion, we have carried out Hall effect measurement at 300 K on a single layer 4 unit cell LaNiO_3 film (Fig. R2.2a) and a PZT/4 uc LaNiO_3 FET in the polarization up (on) state (Fig. R2.2b) and polarization down (off) state (Fig. R2.2c). As shown in Fig. R2.2, the Hall resistance ρ_{xy} vs. B reveals hole doping, consistent with previous report for $R\text{NiO}_3$ films. From the slope $R_H = 1/ne$, we deduce the 3D carrier density to be $5.40 \times 10^{22} \text{ cm}^{-3}$ for the single layer film, which is in good agreement with the theoretically predicted value (Ref. 31) and comparable with the reported carrier density of epitaxial NdNiO_3 (Ref. 11) and $\text{Sm}_{0.5}\text{Nd}_{0.5}\text{NiO}_3$ (Ref. 29) thin films. For the PZT/ LaNiO_3 FET, we obtained the carrier density of $5.36 \times 10^{22}/\text{cm}^3$ for the on state and $4.73 \times 10^{22}/\text{cm}^3$ for the off state. From $2P_r = nte$, with t the film thickness, we deduced the remnant polarization field P_r of PZT to be about $76 \mu\text{C}/\text{cm}^2$, which is consistent with the Hall results obtained on PZT/ $\text{Sm}_{0.5}\text{Nd}_{0.5}\text{NiO}_3$ FET (Ref. 5) and P - E loop measurements taken on a PZT/10 nm LaNiO_3 capacitor (Supplementary Fig. 4). These results confirm that the ferroelectric field effect has led to nonvolatile modulation of the carrier density in the $R\text{NiO}_3$ channel.

Fig. R2.2 | a-c, Asymmetrized ρ_{xy} vs. B taken on a single layer 4 uc LaNiO_3 (a), and the on (b) and off (c) states of a PZT/LNO(4) FET with linear fits (dashe lines). The insets show the $R_{xy}(B)$ data. This figure is included as Supplementary Fig. 6.

To clarify this point, we added Fig. R2.2 and the associated discussion to the Supplementary Note 2 and included the following discussion in the second paragraph on Page 7:

“Hall effect measurements taken on this sample reveal hole type doping and n_{\square} of $8.04 \times 10^{15} \text{ cm}^{-2}$ for the on state and $7.10 \times 10^{15} \text{ cm}^{-2}$ for the off state (Supplementary Note 2), confirming the nonvolatile ferroelectric polarization doping. The on-state carrier density is comparable with that of single layer LaNiO_3 (Supplementary Fig. 6) and similar to those in NdNiO_3 ¹¹ and $\text{Sm}_{0.5}\text{Nd}_{0.5}\text{NiO}_3$ ²⁹ films. From $2P_r = n_{\square}e$, we deduce the remnant polarization P_r of PZT to be about $76 \text{ } \mu\text{C}/\text{cm}^2$, which is in good agreement with P - E loop measurements taken on a PZT/10 nm LaNiO_3 capacitor device ($\sim 77 \text{ } \mu\text{C}/\text{cm}^2$) (Supplementary Fig. 4) and the Hall results obtained on PZT/ $\text{Sm}_{0.5}\text{Nd}_{0.5}\text{NiO}_3$ ⁵.”

Comment 2.9:

“9. Similarly the authors discuss the charge transfer at the LSMO/nickelate interface from literature results. It would be very interesting to measure the Ni and Mn valence through EELS in the STEM specimens. Or alternatively to perform XPS or XAS. This information would also strengthen the author's claims on the underlying mechanisms (and confirm the DFT results).”

Reply 2.9:

We thank the reviewer for raising this important point. Following the reviewer's suggestion, we have analyzed the Mn valence state via EELS Mn L spectra (Fig. R2.3a-c), from which we quantized the L_3 peak position and L_3/L_2 intensity ratio via Lorentzian fitting (Fig. R2.3d). As shown in Fig. R2.4, we observed an enhanced blue shift of the L_3 peak position and suppressed L_3/L_2 ratio towards the $\text{LaNiO}_3/\text{LSMO}$ interface, which can be attributed to the formal valence change from Mn^{3+} to Mn^{4+} . This result clearly reveals the electron transfer from Mn to Ni ions (or hole transfer from Ni to Mn). The change occurs predominately within 1-2 unit cells at the interface with LaNiO_3 , in good agreement with the theoretical predication (Fig. 4e). The L_3 peak position shifts by about 0.7 eV, which corresponds to about 0.06 electron/Mn. This value is slightly smaller than previously reported value for $\text{Sm}_{0.5}\text{Nd}_{0.5}\text{NiO}_3/\text{LSMO}$ heterostructure¹², also consistent with the calculated trend. We also note that the Mn L spectra are observed within 2 uc above and below the interfaces of LSMO (Fig. R2.3b,c). In addition to the intermixing across the interface (Mn diffusion) and atomic scale interface terraces, core-loss signal delocalization, electron probe dechanneling due to the atomic distortion near/at the interface, and multiple scattering can also significantly contribute to the broadening of the EELS signal (Kimoto *et al.*, *Nature* **450**, 702 (2007)).

To clarify this point, we added Fig. R2.3 and the associated discussion to the Supplementary Note 1, added Fig. R2.4 as Fig. 4g in the main text, and included the following discussion in the second paragraph on Page 14:

“We further investigate the charge transfer effect in the PZT/LNO(3)/LSMO(3) sample by analyzing the EELS Mn L spectra (Supplementary Fig. 3). As shown in Fig. 4f, we observe an enhanced blue shift of the L_3 peak position and suppressed L_3/L_2 intensity ratio towards the $\text{LaNiO}_3/\text{LSMO}$ interface, which can be attributed to the formal valence change from Mn^{3+} to Mn^{4+} . This result clearly reveals the electron transfer from Mn to Ni ions (or hole transfer from Ni to Mn). The change occurs predominately within 1-2 unit cells at the interface with LaNiO_3 , in good agreement with the theoretical predication (Fig. 4e). The L_3 peak position shifts by about 0.7 eV, which corresponds to about 0.06 electron/Mn. This value is slightly smaller than previously reported value for $\text{Sm}_{0.5}\text{Nd}_{0.5}\text{NiO}_3/\text{LSMO}$ heterostructure¹², consistent with the calculated trend. Both the charge transfer amount and the length scale of valence distribution are in good agreement with previous reports^{12, 15, 39-41}.”

Fig. R2.3 | EELS Mn spectra. a-c, Cross-sectional HRSTEM image (a), EELS mapping of Mn element (b), and Mn L edge spectra (c) taken on a PZT/LNO(3)/LSMO(3) heterostructure. d, One example of Lorentzian fitting to Mn L_{2,3} peaks of unit cell #9. This figure is included as Supplementary Fig. 3.

Fig. R2.4 | Mn L₃ peak position (top) and L₃/L₂ intensity ratio as a function of unit cell deduced from Fig. R2.3c. The shadowed area highlights possible charge transfer region and serves as a guide to the eye.

Comment 2.10:

“10. More quantitative information on the retention time and fatigue, especially for the best devices, in comparison with the literature.”

Reply 2.10:

Following the reviewer’s suggestion, we have characterized the retention and cycling behaviors of a PZT/LNO(3)/LSMO(2) sample. Figure R2.5(a) shows the time dependence of the normalized on- and off-state resistance. Both polarization states show an initial relaxation, with the resistance values saturating after 200 seconds. Within 20 seconds, the off-state resistance rapidly drops to about 70% of the initial value. In contrast, the on-state resistance only increases about 10%. The relaxation time and saturation level are comparable with previously reported values for (Sm,Nd)NiO₃ single layer channel (Ref. 5) and (Sm,Nd)NiO₃/LSMO bilayer channel (Ref. 12) devices, despite the significantly higher resistance on/off ratio observed in this sample.

To clarify this point, we included Fig. R2.5a as Fig. 3f in the main text and added the following discussion in the third paragraph on Page 10:

“Figure 3f shows the time-dependence of the normalized on- and off-state resistance, defined as $R_{\text{norm}} = [R(t) - R_{\text{on}}] / [R_{\text{off}} - R_{\text{on}}]$, taken on a PZT/LNO(3)/LSMO(2) sample. Resistance in both polarization states shows an initial relaxation and saturates after about 200 seconds. Within the first 20 seconds, the off-state resistance rapidly drops to about 70% of the initial value. In contrast, the on-state resistance only increases by about 10%. This result clear shows that the higher carrier density (*i.e.*, on state) can be directly correlated with a lower resistance relaxation, consistent with the depolarization mechanism. The final resistance on/off ratio settles at about 55% of the initial value. The relaxation time and saturation level are comparable with previously reported values for Sm_{0.5}Nd_{0.5}NiO₃ single layer channel⁵ and Sm_{0.5}Nd_{0.5}NiO₃/LSMO bilayer channel¹² devices, despite the significantly higher $r_{\text{off/on}}$ observed in this sample.”

Figure R2.5b shows the cycling behavior of the same sample by applying ± 5 V voltage pulses via a functional generator. We find the normalized resistance on/off ratio $r_{\text{off/on}} = r_{\text{off/on}} / r_{\text{off/on}}(0)$ is stable up to 10^6 cycles and gradually decreases to about 76% of the initial value after 10^8 cycles (Fig. R2.5b). The $r_{\text{off/on}}$ value remains at this level till 10^{10} cycles. This is consistent with the characteristic three-stage fatigue behavior, *i.e.*, (I) slow fatigue stage, (II) logarithmic stage, and (III) saturated stage (Fig. R2.5c, Ref. 38: Lou, *JAP* **105**, 024101 (2009)). The endurance and saturated values, on the other hand, well surpass devices based on polycrystalline PZT films (Fig. R2.5c, Paton *et al.*, *Integr. Ferroelectr.* **18**, 29 (1997)), supporting the scenario that the film-electrode interface plays a critical role in determining the fatigue behavior (Ref. 38). This material system is thus highly competitive for device applications.

To clarify this point, we added Fig. R2.5b as Fig. 3g in the main text and added the following discussion in the second paragraph on Page 11:

“Next, we examine the cycling behavior of the sample by applying ± 5 V voltage pulses via a functional generator. As shown in Fig. 3g, the resistance on/off ratio is stable up to 10^6 cycles and gradually decreases to about 76% of initial value after 10^8 cycles. It then remains at this level till 10^{10} cycles. This is consistent with the characteristic three stage fatigue behavior, *i.e.*, (I) slow fatigue stage, (II) logarithmic stage, and (III) saturated stage³⁸. The endurance and saturation value, on the other hand, well surpass devices based on polycrystalline PZT films, supporting the scenario that the film-electrode interface plays a critical role in determining the fatigue behavior³⁸. The robust retention and cycling behavior of these Mott FeFETs makes them highly competitive for nanoelectronic applications.”

Fig. R2.5 | **a**, Time-dependence of the normalized on- and off-state resistance, and **b**, cycling test taken on a PZT/LNO(3)/LSMO(2) sample. The initial on state value (defined as 0) is not shown in the logarithmic scale in **(a)**. **c**, Fatigue behavior of a polycrystalline PZT film at different temperatures. Adopted from Paton *et al.*, *Integr. Ferroelectr.* **18**, 29 (1997).

Response to Reviewer 3's Comments

"In this paper, the authors study ferroelectric-gated Mott transistors based on ferroelectric PZT and nickelate / manganite correlated oxides.

Ferroelectric Mott transistors display non-volatile switching and potentially large changes in resistance when switching due to the large (localized) carrier density present in the insulating Mott state. Here the authors report on an extremely large change in resistance when switching the ferroelectric polarization (up to a factor of about 400) in a device combining, for the channel, LaNiO_3 (and NdNiO_3 , SmNiO_3) – a metallic high carrier density material – and $(\text{La,Sr})\text{MnO}_3$ a material known for its colossal magnetoresistance.

This is a detailed complete study on very many samples exploring the role of the thickness of the nickelate compounds and the one of the $(\text{La,Sr})\text{MnO}_3$. The record high resistance switching is obtained when the nickelate layer thickness is close to the thickness at which the material displays a "dead layer" (insulating) behavior – 2.5 unit cells and a LaSrMnO_3 thickness of 2 unit cells.

This is nice paper describing an interesting detailed study of Mott-like transistors using complex oxide heterostructures. The addition of a third LaSrMnO_3 layer in the heterostructure leads to record high resistance switching. The paper can be considered for publication in Nature communications. I have however several questions / comments that the authors should address."

Comment 3.1a:

"I start with an important comment / question on the physics of the studied devices. No doubt that the results are spectacular but there are a few things I do not understand. A Mott transistor is a device in which the channel is in a Mott insulating state (say $1e^-$ per unit cell). The carrier density is extremely high as stated by the authors and the interest is that, under the gate applied electric field, one is moving away from half filling (the Mott state) producing a deconfinement and then conduction takes place with all the electrons that were localized (not only the ones coming from the field effect). So one can restore conduction with a change in carrier density much smaller than the carrier density of the Mott state – unlike what is stated p. 3.

Noticing that NdNiO_3 and SmNiO_3 are thought to be size selective Mott insulators (below about 400K for SmNiO_3 and 200K for NdNiO_3) and LaNiO_3 a paramagnetic metal – not a Mott insulator. My questions are:

Regarding the mechanism proposed and the role of the $(\text{La,Sr})\text{MnO}_3$, the authors say that a charge transfer occurs reducing the carrier density of the RENiO_3 layer. The idea is, as I understand, that this lowering of the carrier density reduces the sheet carrier density boosting the ferroelectric field effect without producing a parallel conduction (low conductivity in $(\text{La,Sr})\text{MnO}_3$).

I can understand the reasoning but not the link to a Mott transistor. If a charge transfer occurs, reducing the RENiO_3 charge density, the system should move away from the Mott state and thus restore conduction (the mechanism of the Mott transistor). With $(\text{La,Sr})\text{MnO}_3$, are the authors claiming that the NdNiO_3 (which is not in a Mott state at RT) and SmNiO_3 layers are in a Mott insulating state at RT?

Reply 3.1a:

We thank the reviewer for the positive comments and valuable suggestions, which help us improve the technical strength and clarity of the paper. First, we'd like to clarify the difference between a Mott transistor and a Mott metal-insulator transition (MIT). We agree with the reviewer's assessment of Mott transition involving the transition from an extended (metallic) electronic state to a localized (insulating) state induced by the change of correlation energy, which can be tuned by pressure or carrier density. For a Mott transistor, on the other hand, we adopt the broad classification that the channel material is a Mott insulator. This device

concept capitalizes on the nonlinear relation between conductivity (σ) vs. carrier density n of a Mott insulator to achieve large field effect modulation: without correlation effect, a metal exhibits a linear σ vs. n relation depicted by the Drude model $\sigma = ne\mu$, with μ the charge mobility; with strong correlation, small change in the carrier density can lead to substantial change in σ and even induce a MIT, *e.g.*, driving the sample across the mobility edge. In this sense, the exact state of the Mott channel is not important. Of course, the Mott transistor exhibit the optimal field effect when the channel is modulated in the vicinity of the MIT. This is indeed what we have observed. As shown in Fig. R3.1, we have tuned a 4 uc LaNiO₃ channel between the metallic and insulating states via the ferroelectric field effect.

Fig. R3.1 | R_{\square} vs T taken on a 4 uc LaNiO₃ channel in the P_{up} and P_{down} states of PZT.

Second, we'd like to emphasize that the field effect tuning in the Mott channel is most effective within the screening length and decays exponentially away from the interface with the PZT gate. Previous studies have shown that the screening length can be on the order of a couple of unit cells (*e.g.*, Refs. 12, 24). This means the bottom region far away from the gate is barely affected by the field effect and can serve as a shunting resistor, reducing the overall resistance modulation. Our proposed material scheme aims at mitigating the effect of this bottom region. With the charge transfer effect, we can effectively reduce the carrier density of the bottom region, making it highly insulating and reducing its contribution to parallel conduction. Our theoretical results (Fig. 4d,e) and EELS data (Fig. 4f) show that the charge transfer length is also on the order of a couple of unit cells, so that the active Mott channel (close to the PZT interface) is not affected by the charge transfer effect.

Reply 3.1b:

LaNiO₃ is not a Mott insulator – it is metallic compound. Are the authors suggesting that ultrathin LaNiO₃ layers, at the verge of the insulating state, are in a Mott insulating state?"

Reply 3.1b:

Thanks for bringing up this important point. Yes, we believe ultrathin LaNiO₃ films are Mott insulators. Previous theoretical studies have predicted that the surface layer of ultrathin LaNiO₃ is an orbital-selective Mott insulator, with the $d_{3z^2-r^2}$ orbitals in the Mott insulating state (Ref. 32: Golalikhani *et al.*, Nat. Commun. **9**, 2206 ((2018)). *Ab initio* calculation also shows the carrier density of ultrathin LaNiO₃ does not exhibit a linear scaling with film thickness but rather drops abruptly at about 2 uc, suggesting that it

approaches the correlation gap (Ref. 31: Fowlie et al., *Adv. Mater.* **29**, 1605197 (2017)). As shown in Fig. R3.1, upon polarization switching of PZT, 4 uc LaNiO₃ exhibits a carrier density driven MIT, which is the hallmark behavior of a Mott insulator. Optical spectroscopy studies of (LaNiO₃)_n/(LaMnO₃)₂ superlattices further show that a MIT is induced in ultrathin LaNiO₃ layers ($n < 5$) due to the charge transfer with LaMnO₃ (Pietro *et al.*, *PRL* **114**, 156801 (2015)). These theoretical and experimental results yield strong support for the scenario that ultrathin LaNiO₃ hosts a Mott insulator state.

To clarify this point, we included Fig. R3.1 as Fig. 3a and added the following discussions to the main text:

Page 4, third paragraph: “Bulk SmNiO₃ and NdNiO₃ are the charge-transfer-type Mott insulators³⁴, while LaNiO₃ is a correlated metal. Epitaxial LaNiO₃ films exhibit a film thickness driven metal-insulator transition upon the dimensionality-crossover²⁸. Previous theoretical studies have predicted that the surface layer of ultrathin LaNiO₃ is an orbital-specific Mott insulator³². *Ab initio* calculations also show the carrier density of ultrathin LaNiO₃ does not scale linearly with film thickness but rather drops abruptly below 5 uc³¹, suggesting that it approaches the correlation gap.”

Page 7, second paragraph: “Next, we investigate the ferroelectric field effect in single layer RNiO₃ channel devices (Fig. 2e). Figure 3a shows $R_{\square}(T)$ taken on a PZT/4 uc LaNiO₃ sample. In the P_{up} state, the sample exhibits metallic behavior at high temperature followed by a resistance upturn at 70 K. By switching the polarization to the P_{down} state, the sample becomes insulating at room temperature, signaling a carrier density driven Mott transition.”

Comment 3.2:

“2. I do not understand the comment bottom of p.3 : “... how to satisfy the requirement of reducing the effective sheet carrier density while preserving screening?” – this requirement is in principle satisfied with a Mott transistor, the change in carrier density to produce conduction being much smaller than the total carrier density as already mentioned above.”

Reply 3.2:

Sorry for the confusing statement. We'd like to clarify that screening here referred to the compensation of depolarization field in PZT in the off state. It is known that the low carrier density renders incomplete screening of the ferroelectric polarization, which is the major factor that causes poor retention in FeFETs (e.g., see Hoffman *et al.*, *Adv. Mater.* **22**, 2957 (2010)). To clarify this point, we modified the first paragraph on Page 4 as follows:

“The central challenge for designing the single layer Mott channel is how to satisfy the conflicting requirements of reducing the effective sheet carrier density while preserving effective screening of the depolarization field in the ferroelectric gate.”

Comment 3.3:

“3. LaNiO₃ is rhombohedral while, at room temperature, NdNiO₃ and SmNiO₃ are orthorhombic, this might be worth mentioning.”

Reply 3.3:

Thanks. We added the following information to the Supplementary Note 1.

“For bulk samples, LaNiO₃ has the rhombohedral structure ($R\bar{3}c$), while NdNiO₃ and SmNiO₃ are orthorhombic (pbm) in the metallic phase¹.”

Comment 3.4:

“4. The authors mention a very high polarization for PZT, $80\mu\text{C}/\text{cm}^2$. Was this measured? What is the estimate of the polarization obtained from the resistance switching (obtained from the change in sheet carrier density)?”

Reply 3.4:

We characterized the polarization of PZT from the P - E loop and Hall effect measurements. Figure R3.2a shows the P - E loop taken on an Au/PZT/10 nm LaNiO₃ capacitor device. We apply a triangular wave of bias voltage to the capacitor and measure the switching current as a function of time. The polarization is calculated by the time integral of the switching current. From the hysteresis, we obtain a remanent polarization P_r of $77\mu\text{C}/\text{cm}^2$ for PZT. We also carried out Hall effect measurement at 300 K on the PZT/4 nm LaNiO₃ FET in the polarization up (on) state (Fig. R3.2b) and polarization (off) state (Fig. R3.2c). The Hall resistance ρ_{xy} vs B reveals hole doping, consistent with previous report. From the slope $R_H = 1/ne$, we obtained the carrier density of $5.36 \times 10^{22}\text{ cm}^{-3}$ for the on state and $4.73 \times 10^{22}\text{ cm}^{-3}$ for the off state. From $2P_r = n_{\square}e = nte$, with t the film thickness, we deduced P_r of PZT to be about $76\mu\text{C}/\text{cm}^2$, consistent with the value from the P - E loop measurement. To clarify this point, we added Fig. R3.2 and the associated discussion to the Supplementary Note 2 and included the following discussion in the second paragraph on Page 7:

“Hall effect measurements taken on this sample reveal hole type doping and n_{\square} of $7.10 \times 10^{15}\text{ cm}^{-2}$ for the off state and $8.04 \times 10^{15}\text{ cm}^{-2}$ for the on state (Supplementary Note 2), confirming the nonvolatile ferroelectric polarization doping. The on state carrier density is comparable with that of single layer LNO (Supplementary Fig. 6) and similar to those in NNO¹¹ and SNNO²⁹ films. From $2P_r = n_{\square}e$, we deduce the remnant polarization P_r of PZT to be about $76\mu\text{C}/\text{cm}^2$, which is consistent with the Hall results obtained on PZT/SNNO FET⁵ and P - E loop measurements taken on a PZT/10 nm LNO capacitor device ($\sim 77\mu\text{C}/\text{cm}^2$) (Supplementary Fig. 4).”

Fig. R3.2. | **a**, P - E loop taken on an Au/PZT/10 nm LaNiO₃ capacitor. This is included as the Supplementary Fig. 4. **b,c**, Asymmetrical ρ_{xy} vs. B taken on the on (b) and off (c) states of a PZT/LNO(4) FET with linear fits (dashed lines). The insets show the raw $R_{xy}(B)$ data. This is included as the Supplementary Fig. 6.

Comment 3.5:

“5. Have the authors characterize the behavior of single layers of LaSrMnO₃? What is the sheet resistance of the LaSrMnO₃ layer?”

Reply 3.5:

We characterized the four-point resistance of a 2 μm LSMO deposited on SrTiO_3 substrate. The sheet resistance is about 530 $\text{M}\Omega$ at 300 K, which is orders of magnitude higher than that of RNiO_3 films at the thickness range for the FET devices (Fig. 2). We now added this information to the second paragraph on Page 10:

“As ultrathin LSMO is too insulating ($R_{\square} \sim 530 \text{ M}\Omega$ for 2 μm LSMO), it cannot provide sufficient parallel conduction to compensate the reduced conduction in LaNiO_3 due to the charge transfer effect.”

Comment 3.6:

“6. P.4 a detail – channel not chaannel.”

Reply 3.6:

Thanks. We corrected this typo.

List of Changes

Main Text:

1. Title: “Record High Room Temperature Resistance Switching in Ferroelectric-Gated Mott Transistors Unlocked by Interfacial Charge Engineering”
2. Abstract: “The superior size and power scaling potential of ferroelectric-gated Mott transistors makes them promising building blocks for developing energy-efficient memory and logic applications in the post-Moore’s Law era. The close to metallic carrier density in the Mott channel, however, imposes the bottleneck for achieving substantial field effect modulation via a solid-state gate.”
3. Page 4, first paragraph: “The central challenge for designing the single layer Mott channel is how to satisfy the conflicting requirements of reducing the effective sheet carrier density while preserving effective screening of the depolarization field in the ferroelectric gate.”
4. Page 4, third paragraph: “Bulk SmNiO_3 and NdNiO_3 are the charge-transfer-type Mott insulators³⁴, while LaNiO_3 is a correlated metal. Epitaxial LaNiO_3 films exhibit a film thickness driven metal-insulator transition upon the dimensionality-crossover²⁸. Previous theoretical studies have predicted that the surface layer of ultrathin LaNiO_3 is an orbital-specific Mott insulator³². *Ab initio* calculations also show the carrier density of ultrathin LaNiO_3 does not scale linearly with film thickness but rather drops abruptly below 5 uc ³¹, suggesting that it approaches the correlation gap.”
5. Page 7, second paragraph: “Next, we investigate the ferroelectric field effect in single layer RNiO_3 channel devices (Fig. 2e). Figure 3a shows $R_{\square}(T)$ taken on a PZT/4 uc LaNiO_3 sample. In the P_{up} state, the sample exhibits metallic behavior at high temperature followed by a resistance upturn at 70 K. By switching the polarization to the P_{down} state, the sample becomes insulating at room temperature, signaling a carrier density driven Mott transition. Hall effect measurements taken on this sample reveal hole type doping and n_{\square} of $8.04 \times 10^{15} \text{ cm}^{-2}$ for the on state and $7.10 \times 10^{15} \text{ cm}^{-2}$ for the off state (Supplementary Note 2), confirming the nonvolatile ferroelectric polarization doping. The on-state carrier density is comparable with that of single layer LaNiO_3 (Supplementary Fig. 6) and similar to those in NdNiO_3 ¹¹ and $\text{Sm}_{0.5}\text{Nd}_{0.5}\text{NiO}_3$ ²⁹ films. From $2P_r = n_{\square}e$, we deduce the remnant polarization P_r of PZT to be about $76 \mu\text{C}/\text{cm}^2$, which is in good agreement with P - E loop measurements taken on a PZT/10 nm LaNiO_3 capacitor device ($\sim 77 \mu\text{C}/\text{cm}^2$) (Supplementary Fig. 4) and the Hall results obtained on PZT/ $\text{Sm}_{0.5}\text{Nd}_{0.5}\text{NiO}_3$ ⁵.”
6. Page 7, third paragraph: “We also note that depositing the PZT top-layer suppresses the channel conductivity. The on-state R_{\square} of PZT-gated single layer RNiO_3 channels is always higher than that of the uncapped film with the same thickness. Similar suppression in conduction has also been observed in the NdNiO_3 and SmNdO_3 channels. Such a change may share a similar mechanism as that observed in Ref. [35], where a closely lattice matched capping layer LaAlO_3 can suppress the lattice distortion induced by surface reconstruction and hence enhance LaNiO_3 conductivity. The PZT capping layer, in contrast, imposes a large tensile strain, which suppresses the RNiO_3 conduction^{29,30}.”
7. Page 10, second paragraph: “As ultrathin LSMO is too insulating ($R_{\square} \sim 530 \text{ M}\Omega$ for 2 uc LSMO), it cannot provide sufficient parallel conduction to compensate the reduced conduction in LaNiO_3 due to the charge transfer effect.”
8. Page 10, third paragraph: “Figure 3f shows the time-dependence of the normalized on- and off-state resistance, defined as $R_{\text{norm}} = [R(t) - R_{\text{on}}]/[R_{\text{off}} - R_{\text{on}}]$, taken on a PZT/LNO(3)/LSMO(2) sample. Resistance in both polarization states shows an initial relaxation and saturates after about 200 seconds. Within the first 20 seconds, the off-state resistance rapidly drops to about 70% of the initial value. In

contrast, the on-state resistance only increases by about 10%. This result clearly shows that the higher carrier density (*i.e.*, on state) can be directly correlated with a lower resistance relaxation, consistent with the depolarization mechanism. The final resistance on/off ratio settles at about 55% of the initial value. The relaxation time and saturation level are comparable with previously reported values for $\text{Sm}_{0.5}\text{Nd}_{0.5}\text{NiO}_3$ single layer channel⁵ and $\text{Sm}_{0.5}\text{Nd}_{0.5}\text{NiO}_3/\text{LSMO}$ bilayer channel¹² devices, despite the significantly higher $r_{\text{off/on}}$ observed in this sample.”

9. Page 11, second paragraph: “Next, we examine the cycling behavior of the sample by applying ± 5 V voltage pulses via a function generator. As shown in Fig. 3g, the resistance on/off ratio is stable up to 10^6 cycles and gradually decreases to about 76% of initial value after 10^8 cycles. It then remains at this level till 10^{10} cycles. This is consistent with the characteristic three stage fatigue behavior, *i.e.*, (I) slow fatigue stage, (II) logarithmic stage, and (III) saturated stage³⁸. The endurance and saturation value, on the other hand, well surpass devices based on polycrystalline PZT films, supporting the scenario that the film-electrode interface plays a critical role in determining the fatigue behavior³⁸. The robust retention and cycling behavior of these Mott FeFETs makes them highly competitive for nanoelectronic applications.
10. Page 12, third paragraph: “In previous studies, the charge transfer effect has been intensively investigated in various nickelate/manganite heterointerfaces, including the $\text{LaNiO}_3/\text{La}_{2/3}\text{Sr}_{1/3}\text{MnO}_3$ ³⁹ and $\text{LaNiO}_3/\text{LaMnO}_3$ (LMO)^{15, 40, 41} superlattices and $\text{Sm}_{0.5}\text{Nd}_{0.5}\text{NiO}_3/\text{La}_{0.67}\text{Mn}_{0.33}\text{MnO}_3$ ¹² and $\text{La}_{0.7}\text{Sr}_{0.3}\text{MnO}_3/\text{NdNiO}_3$ ⁴² heterostructures. In these studies, x-ray adsorption spectroscopy studies show a consistent blue shift of Mn L_3 edge by 0.7-1.15 eV^{12, 15, 40, 41}, corresponding to a nominal valence change of $\text{Ni}^{3+} + \text{Mn}^{3+} \rightarrow \text{Ni}^{2+} + \text{Mn}^{4+}$ by about 0.06-0.1 electron/Mn. The charge coupling has led to emergent antiferromagnetic/ferromagnetic states^{15, 40, 42}, noncolinear magnetic structure³⁹, and interfacial exchange coupling⁴². The reported L_{ct} varies from 1-4 monolayers^{12, 15, 39-41}.”
11. Page 14, first paragraph: “Reducing the Sr content, *e.g.*, replacing LSMO with LaMnO_3 , can also lead to larger charge transfer.”
12. Page 14, second paragraph: “We further investigate the charge transfer effect in the PZT/LNO(3)/LSMO(3) sample by analyzing the EELS Mn L spectra (Supplementary Fig. 3). As shown in Fig. 4f, we observe an enhanced blue shift of the L_3 peak position and suppressed L_3/L_2 intensity ratio towards the $\text{LaNiO}_3/\text{LSMO}$ interface, which can be attributed to the formal valence change from Mn^{3+} to Mn^{4+} . This result clearly reveals the electron transfer from Mn to Ni ions (or hole transfer from Ni to Mn). The change occurs predominately within 1-2 unit cells at the interface with LaNiO_3 , in good agreement with the theoretical prediction (Fig. 4e). The L_3 peak position shifts by about 0.7 eV, which corresponds to about 0.06 electron/Mn. This value is slightly smaller than previously reported value for $\text{Sm}_{0.5}\text{Nd}_{0.5}\text{NiO}_3/\text{LSMO}$ heterostructure¹², consistent with the calculated trend. Both the charge transfer amount and the length scale of valence distribution are in good agreement with previous reports^{12, 15, 39-41}.”
13. Page 16, second paragraph: “These devices show superior retention and cycling behavior.”
14. Page 16, third paragraph: “The channel size varies from $80 \times 40 \mu\text{m}^2$ to $10 \times 5 \mu\text{m}^2$.”
15. Page 17, first paragraph: “The detailed device fabrication process is discussed in the Supplementary Note 2.”
16. Page 17, third paragraph: “For the Mn valence state analysis, the energy resolution was ~ 0.7 eV based on measured full width at half maximum (FWHM) of the zero-loss peak with 0.09 eV/channel.”

Figures:

17. Figure 2 caption: “ $R_{\square}(T)$ of (a) LaNiO_3 , (b) NdNiO_3 , and (c) SmNiO_3 films with various thicknesses deposited on LaAlO_3 substrates.”
18. Figure 3: We included $R_{\square}(T)$ for the on- and off- state of a PZT/LNO(4) sample as Fig. 3a and added the associated caption.
19. Figure 3: We included the retention and cycling data for a PZT/LNO(3)/LSMO(2) sample as Fig. 3e,f, respectively, and added the associated caption.
20. Figure 4: We added EELS Mn L edge data as Fig. 4f and added the associated caption.

Supplementary Information:

21. Supplementary Note 1: We added discussions of crystal structures of LaNiO_3 , NdNiO_3 , and SmNdO_3 and the EELS data taken on a PZT/LNO(3)/LSMO(3) sample, and included the corresponding figure as Supplementary Fig. 3.
22. Supplementary Note 2: We added a new Note on device fabrication and characterization.
23. We included a new figure of DFT+U calculation of $\text{NdNiO}_3(4)/\text{LaMnO}_3(4)$ superlattice as Supplementary Fig. 12b and added associated discussion.

REVIEWERS' COMMENTS

Reviewer #1 (Remarks to the Author):

In the revised manuscript (now) entitled "Record high room temperature resistance switching on ferroelectric-gated Mott transistors unlocked by interfacial charge engineering" by Y. Hao et al., the authors adequately addressed the minor comments I had regarding the prior version, as well as the issues raised by the other referees. I fully support publication of the manuscript in Nature Communications for the reasons outlined in my previous report.

There is a typo in the revised manuscript. "absorption" is misspelled on the bottom of page 12. "In these studies, x-ray adsorption spectroscopy studies show a consistent blue shift. . . "

Reviewer #2 (Remarks to the Author):

I think the Authors have done a very good job in addressing my comments and those of the other referees. I can now gladly recommend this excellent paper for publication in Nature Comms.

Reviewer #3 (Remarks to the Author):

Second referee report

I read the reply of the authors. They answered to my questions and comments.

A key point is that the authors consider that the ultrathin nickelate layers used are in a Mott insulating state. I am not sure it has been demonstrated that insulating ultrathin LaNiO₃ films are in a Mott state but it is true that, for SrVO₃, ARPES experiments show that ultrathin films turn into such a state.

I do not agree with the authors when they are saying “the exact state of the Mott channel is not important”. The concept of the Mott transistor is quite well defined and it relies on leaving the Mott state as the carrier density is increased or decreased from half filling. So one is tuning n not U/W – so the tuning is not modifying the correlation energy as stated in the reply.

This said, I think the paper reports on very interesting results and can be published.

Manuscript NCOMMS-23-23990-A: Response to Reviewers' Comments

Reviewer #1's Comments:

"In the revised manuscript (now) entitled "Record high room temperature resistance switching on ferroelectric-gated Mott transistors unlocked by interfacial charge engineering" by Y. Hao et al., the authors adequately addressed the minor comments I had regarding the prior version, as well as the issues raised by the other referees. I fully support publication of the manuscript in Nature Communications for the reasons outlined in my previous report.

1. There is a typo in the revised manuscript. "absorption" is misspelled on the bottom of page 12. "In these studies, x-ray adsorption spectroscopy studies show a consistent blue shift. . ."

Reply:

We thank the reviewer for reviewing our manuscript and appreciate the recommendation for publication. We have corrected the typo in the main text.

Reviewer 2's Comments:

"I think the Authors have done a very good job in addressing my comments and those of the other referees. I can now gladly recommend this excellent paper for publication in Nature Comms.

Reply:

We thank the reviewer for reviewing our manuscript and appreciate the recommendation for publication.

Reviewer 3's Comments

"I read the reply of the authors. They answered to my questions and comments.

A key point is that the authors consider that the ultrathin nickelate layers used are in a Mott insulating state. I am not sure it has been demonstrated that insulating ultrathin LaNiO₃ films are in a Mott state but it is true that, for SrVO₃, ARPES experiments show that ultrathin films turn into such a state.

I do not agree with the authors when they are saying "the exact state of the Mott channel is not important". The concept of the Mott transistor is quite well defined and it relies on leaving the Mott state as the carrier density is increased or decreased from half filling. So one is tuning n not U/W – so the tuning is not modifying the correlation energy as stated in the reply.

This said, I think the paper reports on very interesting results and can be published."

Reply 3.1:

We thank the reviewer for reviewing our manuscript and appreciate the recommendation for publication. We'd like to clarify that we fully agree with the reviewer that the Mott transistor operates on the tuning of carrier density, not U/W . What we meant is this device concept capitalizes on the highly nonlinear σ vs n relation in the vicinity of the metal-insulator transition (MIT). To achieve a large resistance modulation, whether the exact starting (unmodulated) state of the Mott channel is in the insulating or metallic phase is not important as long as it is close to the MIT.